# RETRO SYNFLOW: Discrete Flow Matching for Accurate and Diverse Single-Step Retrosynthesis

**Robin Yadav**[1]
robiny12@student.ubc.ca

**Qi Yan**[1,2]
qi.yan@ece.ubc.ca

**Guy Wolf**[3,4,5]
wolfguy@mila.quebec

**Avishek Joey Bose**[3,6]
joey.bose@mail.mcgill.ca

**Renjie Liao**[1,2,5]
rjliao@ece.ubc.ca

[1]UBC; [2]Vector Institute; [3]Mila; [4]Université de Montréal;
[5]Canada CIFAR AI Chair; [6]University of Oxford

## Abstract

A fundamental problem in organic chemistry is identifying and predicting the series of reactions that synthesize a desired target product molecule. Due to the combinatorial nature of the chemical search space, single-step reactant prediction—*i.e.* single-step retrosynthesis—remains challenging even for existing state-of-the-art template-free generative approaches to produce an accurate yet diverse set of feasible reactions. In this paper, we model single-step retrosynthesis planning and introduce RETRO SYNFLOW (RSF) a discrete flow-matching framework that builds a Markov bridge between the prescribed target product molecule and the reactant molecule. In contrast to past approaches, RSF employs a reaction center identification step to produce intermediate structures known as synthons as a more informative source distribution for the discrete flow. To further enhance diversity and feasibility of generated samples, we employ Feynman-Kac steering with Sequential Monte Carlo based resampling to steer promising generations at inference using a new reward oracle that relies on a forward-synthesis model. Empirically, we demonstrate RSF achieves $60.0\%$ top-1 accuracy, which outperforms the previous SOTA by $20\%$. We also substantiate the benefits of steering at inference and demonstrate that FK-steering improves top-5 round-trip accuracy by $19\%$ over prior template-free SOTA methods, all while preserving competitive top-$k$ accuracy results.

## 1 Introduction

Retrosynthesis planning is a fundamental problem in chemistry that involves decomposing a complex target molecule (the product) into simpler, commercially available structures (the reactants) to establish synthesis routes [8, 45]. This process is crucial for verifying the synthesizability of proposed molecules with desirable properties, particularly in drug discovery [43]. For instance, retrosynthesis is critical for lead optimization in medicinal chemistry, which requires designing efficient synthetic routes to modify chemical structures to enhance a compound's potency, selectivity, and pharmacokinetic properties [25]. Traditionally, chemists manually identify and validate reactants and pathways, a labor-intensive process exacerbated by the vast search space of transformations from reactant to product molecules. This enduring challenge has driven decades of research in computer-assisted retrosynthesis [8], with recent advances in machine learning (ML) enabling more effective exploration of the combinatorial reactant space [3, 6, 48, 9]. Such methods are promising in significantly accelerate the drug discovery pipeline.

The current dominant paradigm for ML-based retrosynthesis planning consists of two main components: a *single-step retrosynthesis* model and a multi-step planning algorithm. These ML-based approaches can be broadly categorized as *template-based* and *template-free* methods. Template-based

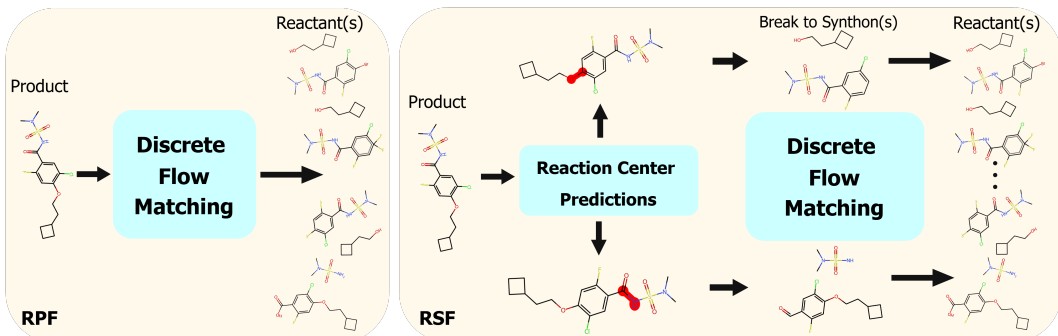

Figure 1: An overview of our RETRO PRODFLOW (RPF) and RETRO SYNFLOW (RSF) framework. RPF directly maps a product molecule to reactants via discrete flow. RSF first predicts synthons from the product using a reaction center predictor, then maps these synthons to reactants via discrete flow.

methods rely on a predefined database of reaction templates with hand-crafted specificity [36], which ensures syntactic and chemical validity but often limits diversity and generalization to novel reaction types. In contrast, template-free methods and *semi-template* methods are more flexible and capable of predicting reactions not seen in existing databases [50], offering improved generalization. More recently, template-free and semi-template approaches have emerged as a natural fit for generative modeling techniques, enabling retrosynthesis prediction to be formulated as a *conditional generation problem*: generating reactants conditioned on a given product molecule. Semi-template methods increase interpretability by breaking the generation process into two steps by first identifying intermediate molecular structures called synthons and completing synthons to form reactants. While promising, many existing generative methods rely on sequential molecular representations like SMILES [39], and as a result, fail to capture the rich chemical contexts encoded in molecular attributed graphs.

**Current work**. In this paper, we frame single-step retrosynthesis as learning a transport map from a source distribution to an intractable target data distribution using finite paired samples. We represent molecules as attributed graphs and propose two template-free/semi-template generative models— RETRO PRODFLOW (RPF) and RETRO SYNFLOW (RSF)–based on recent advances in discrete flow matching [14, 2]. As discrete flows are flexible in choosing the source distribution, we explore two options as shown in Figure 1. First, in RPF, we use the product distribution directly as the source and learn a flow that transforms products to reactants. Second, we leverage pretrained reaction center prediction models to construct an informative source distribution over *synthons*—intermediate molecular fragments obtained by decomposing the product at its reaction center.

This reduces the original problem to a simpler conditional generation task, *i.e.*, mapping from synthons to reactants, and improves performance. Both models learn continuous-time Markov chains to stochastically transport product molecules to reactants, benefiting from fast inference and high sample quality. Furhtermore, they naturally support scoring and ranking of generated reactants via its stochastic formulation.

As the space of possible reactants that synthesize into valid products is combinatorially large, a fundamental design goal in retrosynthesis is to generate diverse candidate reactants that simultaneously achieve high accuracy. To address this, we leverage Feynman-Kac (FK) steering [40], an inference-time steering method that guides sampling of flow models towards more feasible and diverse outputs. FK steering employs sequential Monte-Carlo (SMC), a particle-based method

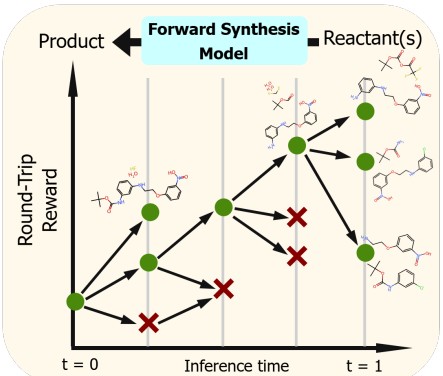

Figure 2: Inference time steering with a forward-synthesis reward model.

that resamples promising intermediate candidates throughout the generation process based on a reward function. We define this reward using a forward-synthesis model to enforce round-trip consistency—a standard measure of diversity and feasibility. We highlight the benefits of reward-based inference time steering, demonstrating a $13\%$ improvement in top-$5$ round-trip accuracy over prior SOTA methods.

In summary, our main contributions are listed below,

1. We propose the first flow matching framework for retrosynthesis, introducing two variants: RETRO PRODFLOW (RPF), which maps products directly to reactants, and RETRO SYNFLOW (RSF), which leverages *synthons* (intermediate molecular fragments) to simplify the generation task.

2. We improve the diversity and feasibility of generated reactants using FK-steering, an inference-time technique guided by a forward-synthesis reward. This yields a 19% gain in top-5 round-trip accuracy over prior template-free methods.

3. We show that using synthons as an inductive prior significantly enhances performance. RSF achieves 60% top-1 accuracy on the USPTO-50k benchmark, outperforming state-of-the-art template-free and semi-template methods.

## 2 Background and preliminaries

**Notations**. Given a vocabulary set $\mathcal{X}$ with $d$ elements, we establish a bijection between $\mathcal{X}$ and the index set $[d] = \{1, \ldots, d\}$. Accordingly, any discrete data $x$ drawn from $\mathcal{X}$ can be represented as an integer index in $[d]$. A categorical distribution over $\mathcal{X}$, denoted $\mathrm{Cat}(x; p)$, is given by $p(x = i) = p^i$, where $\sum_{i=1}^{d} p^i = 1$ and $p^i \geq 0, \forall i$. A sequence $\mathbf{x} = (\mathbf{x}^1, \ldots, \mathbf{x}^n)$ of $n$ tokens is defined over the product space $\mathcal{X}^n$. We assume a dataset of such sequences is sampled from a target data distribution $p_{\mathrm{data}}$. Discrete flow matching models, like their continuous counterparts, aim to transport a source distribution $p_{\mathrm{source}} := p_0$ defined at time $t = 0$ to the data distribution $p_{\mathrm{data}} := p_1$ at time $t = 1$.

### 2.1 Discrete Flow Matching

Discrete flow matching operates directly on discrete data, mirroring the construction of flow matching models over continuous spaces. Analogously, our goal is to construct a generative probability path, $p_t$ that interpolates between the source and target distributions. The key insight of flow matching is to construct $p_t$ by marginalizing simpler probability paths conditioned on samples from the source and data distributions. A conditional probability path, $p_t(\cdot|\mathbf{x}_0, \mathbf{x}_1)$ is a time-evolving distribution satisfying $p_0(\mathbf{x}^i|\mathbf{x}_0, \mathbf{x}_1) = \delta(\mathbf{x}_0^i)$ and $p_1(\mathbf{x}^i|\mathbf{x}_0, \mathbf{x}_1) = \delta(\mathbf{x}_1^i)$ where $\mathbf{x}_0 \sim p_0$ and $\mathbf{x}_1 \sim p_1$. Conditional probability paths are independent across each dimension of the sequence, with the simplest choice being a convex combination of $p_0(\mathbf{x}^i|\mathbf{x}_0, \mathbf{x}_1)$ and $p_1(\mathbf{x}^i|\mathbf{x}_0, \mathbf{x}_1)$,

$$p_t(\mathbf{x}_t^i|\mathbf{x}_0, \mathbf{x}_1) = \mathrm{Cat}(\mathbf{x}_t^i; (1-t)\delta(\mathbf{x}_0^i) + t\delta(\mathbf{x}_1^i)), \quad \text{where } p_t(\mathbf{x}_t|\mathbf{x}_0, \mathbf{x}_1) = \prod_{i=1}^{n} p_t(\mathbf{x}_t^i|\mathbf{x}_0, \mathbf{x}_1). \tag{1}$$

Similar to the continuous setting, discrete flow matching constructs a generating *probability velocity* $u_t(\cdot, \mathbf{x}_t) \in \mathbb{R}^n$, which models the rate of probability mass change of the sample $\mathbf{x}_t$ in each of its $n$ positions. Specifically, we view $(\mathbf{x}_t)_{0 \leq t \leq 1}$ as a collection of random variables that form a continuous-time Markov Chain (CTMC), jumping between states in $\mathcal{X}^n$. Each position of $\mathbf{x}_t$ can be simulated by the following probability transition kernel $p_{t+h|t}(\mathbf{x}_{t+h}^i|\mathbf{x}_t) = \mathrm{Cat}(\mathbf{x}_{t+h}^i; \delta(\mathbf{x}_t^i) + h u_t^i(\mathbf{x}_{t+h}^i, \mathbf{x}_t))$. Thus, sampling from the CTMC and simulating a trajectory from $p_t$ given its velocity $u_t$ is straightforward. We start from a sample $\mathbf{x}_0 \sim p_0$ from the source distribution and update each dimension with the transition kernel $\mathbf{x}_{t+h}^i \sim p_{t+h|t}(\mathbf{x}_{t+h}^i|\mathbf{x}_t)$. This results in samples from the desired data distribution $p_1$. Analogous to continuous flow matching, $u_t^i$ is constructed by marginalizing the conditional probability velocities that generate the conditional probability paths. For the simple conditional probability paths given by Eq. 1, the marginal probability velocity is $u_t^i(\mathbf{x}^i, \mathbf{x}_t) = \left(p_{1|t}(\mathbf{x}^i|\mathbf{x}_t) - \delta(\mathbf{x}_t^i)\right)/(1-t)$. The intractable posterior distribution, $p_{1|t}$, known as the probability *denoiser*, predicts a clean sample $\mathbf{x}_1$ from an intermediate noisy sample $\mathbf{x}_t$. We can approximate the denoiser with a neural network $p_\theta(\mathbf{x}_1^i|\mathbf{x}_t)$ and train it by minimizing a cross-entropy loss that forms a weighted evidence lower bound (ELBO) on $\log p_{1,\theta}(\mathbf{x}_1)$ [12].

### 2.2 Feynman-Kac Steering

Given a trained flow model whose marginal distribution at time $t = 1$ is denoted by $p_\theta(\mathbf{x}_1)$. We are interested in sampling from a target distribution that tilts $p_\theta(\mathbf{x}_1)$ using a terminal (parametrized) reward function $r : \mathcal{X}^n \to [0, 1]$, that consumes fully denoised sequences $\mathbf{x}_1$,

$$p_{\mathrm{target}}(\mathbf{x}_1) = \frac{1}{\mathcal{Z}} p_\theta(\mathbf{x}_1) \exp(\lambda r(\mathbf{x}_1)), \tag{2}$$

where $\lambda$ controls the steering intensity, and $\mathcal{Z}$ is a normalization constant. Sampling trajectories in flow models involves discretizing the interval $[0,1]$ into a grid of timesteps $\{0, h, \cdots, 1-h, 1\}$ and sampling from the learned transition kernel $p_{t+h|t}$. To steer the sampling process toward high-reward outcomes, we employ *Feynman-Kac (FK) steering* [40], which modifies the transition kernels using potential functions that favor trajectories $\tau(\mathbf{x}_{0:1})$ ending with high-reward $\mathbf{x}_1$ samples. The FK process begins from a reweighted initial distribution $p_{\text{FK},0}(\mathbf{x}_0) \propto p_{\text{source}}(\mathbf{x}_0)U_0(\mathbf{x}_0)$ and iteratively build $p_{\text{FK},t+h}$ by tilting the transition kernel with a potential $U_t(\mathbf{x}_0, \cdots, \mathbf{x}_{t-h}, \mathbf{x}_t)$:

$$p_{\text{FK},t+h}(\mathbf{x}_{t+h}) = \frac{1}{\mathcal{Z}_t} p_{t+h|t}(\mathbf{x}_{t+h}|\mathbf{x}_t) U_t(\mathbf{x}_{0:t}) \underbrace{p_\theta(\mathbf{x}_{0:t}) \prod_{s \in \{0, \cdots, t-h, t\}} U_s(\mathbf{x}_{0:s})}_{\propto p_{\text{FK},t}}.$$

Since direct sampling from $p_{\text{FK},1}$ is intractible, we employ Sequential Monte Carlo (SMC) methods. SMC begins with $K$ particles $\{\mathbf{x}_0^m\}_{m=0}^K$ sampled from the source distribution. At each transition step, it updates their importance weights and resamples the particles accordingly. The importance weight for particle $m$ at time $t + h$ is given by $w_{t+h}^m = p_{t+h|t}(\mathbf{x}_{t+h}^m|\mathbf{x}_t)U_t(\mathbf{x}_{0:t}^m)$.

## 3 Discrete Flow Matching for Retrosynthesis

We model single-step retrosynthesis using discrete flow matching by representing product and reactant molecules as a pair of molecular graphs $(G^p, G^r)$. Our focus is on the single product setting, *i.e.*, a single product molecule corresponds to a set of reactant molecules. The reactants are represented as a single disconnected graph to account for multiple molecules. A molecular graph $G = (\mathbf{v}, \boldsymbol{E})$ with $N$ atoms consists of: a node feature vector $\mathbf{v} \in [K_n]^N$, where $\mathbf{v}^i$ encodes the atom type of atom $i$ (out of $K_n$ possible types), and an edge feature matrix $\boldsymbol{E} \in [K_e]^{N \times N}$, where $\boldsymbol{E}^{i,j}$ indicates the bond type (among $K_e$ possible types) between atoms $i$ and $j$.

In this formulation, both $\mathbf{v}$ and $\boldsymbol{E}$ can be viewed as collections of discrete random variables. We aim to learn a generative probability path $p_t$ that interpolates between a source and the data distributions over graphs of product and reactant pairs. Correspondingly, the design of the conditional probability path of a graph, $p_t(G_t|G_0, G_1)$ factorizes over the nodes and edges as follows:

$$p_t(\mathbf{v}_t^i|\mathbf{v}_0, \mathbf{v}_1) = (1-t)\delta(\mathbf{v}_0^i) + t\delta(\mathbf{v}_1^i), \quad p_t(\boldsymbol{E}_t^{i,j}|\boldsymbol{E}_0, \boldsymbol{E}_1) = (1-t)\delta(\boldsymbol{E}_0^{i,j}) + t\delta(\boldsymbol{E}_1^{i,j}).$$

To complete the specification of a discrete flow matching model, we must also define both the source and data distributions. For retrosynthesis, the data distribution, $p_{\text{data}}$ is simply the distribution of reactant molecules. We explore two choices for the source distribution. The most natural option is to set the source distribution, $p_{\text{source}}$ to the empirical distribution of product molecules. This leads to our first model, RETRO PRODFLOW (RPF), which directly transports product molecules to their corresponding reactants. The second option, which we discuss in detail in the following section, sets the source distribution to the space of intermediate synthons which acts as a more informative structural prior. Since some atoms present in the product may not appear in the reactants, we follow RetroBridge [18] and append dummy atoms to the product to ensure alignment of node dimensions. More details are discussed in Appendix A.

The probabilistic formulation of flow matching provides a natural way to select the most likely reactants for a product molecule out of all possible generations. For a set of $N$ samples, $\{\mathbf{x}_1^i\}_{i=1}^N$ generated by the flow matching model for $\mathbf{x}_0 \sim p_{\text{source}}$, the score of sample $\hat{\mathbf{x}}_1$ is the empirical frequency. This score is an estimation of the probability of $\hat{\mathbf{x}}_1$ given $\mathbf{x}_0$, i.e.,

$$p_\theta(\hat{\mathbf{x}}_1|\mathbf{x}_0) = \mathbb{E}_{\mathbf{z} \sim p_\theta(\cdot|\mathbf{x}_0)} \mathbb{1}[\mathbf{z} = \hat{\mathbf{x}}_1] \approx \frac{1}{N} \sum_{i=1}^N \mathbb{1}[\mathbf{x}_1^i = \hat{\mathbf{x}}_1].$$

### 3.1 RETRO SYNFLOW

In our approach, we inject more valuable structural information into the flow matching process by setting the distribution of synthons as a more informative source. This is motivated by the two-stage formulation of single-step retrosynthesis: reaction center identification and synthon completion. The two-stage approach improves interpretability by closely mirroring the way expert chemists reason about retrosynthesis. Synthons are hypothetical intermediate molecules representing potential

reactants [39]. Although a synthon may not correspond to a chemically valid molecule, it can be transformed into one by adding suitable leaving groups that account for reactivity.

Synthon generation begins by identifying a reaction center in the product molecule. A reaction center is defined as a pair of atoms $(i, j)$ in the product satisfying two criteria: 1) atoms $i$ and $j$ are connected by a bond in the product molecule, and 2) there is no bond between atoms $i$ and $j$ in the reactant(s). We can derive the synthon molecule(s) by deleting the bond that connects the atoms in the reaction center.

Our primary focus lies in the second step of retrosynthesis: applying flow matching for synthon completion. In this case, the source distribution $p_{\text{source}}$ is the distribution of synthons. Given a product molecule $G^p$, we use a reaction center prediction model to output $M$ potential reaction-center candidates. This results in $M$ predictions for the set of synthon(s). Each set of synthon(s) is treated as a single disconnected graph, $G^s$. To train the discrete flow model, we once again decompose the conditional probability path $p_t(G_t|G_0, G_1)$ over the nodes and edges where $G_0 := G^s$ and $G_1 := G^r$. Therefore, we can model the generating probability velocity and update the tokens of the nodes and edges separately according to their respective transition kernels:

$$u^i_{\text{nodes},t}(\mathbf{v}^i, \mathbf{v}_t) = \frac{1}{1-t} \left[ p_\theta(\mathbf{v}^i|\mathbf{v}_t, \boldsymbol{E}_t) - \delta(\mathbf{v}^i_t) \right],$$

$$u^{i,j}_{\text{edges},t}(\boldsymbol{E}^{i,j}, \mathbf{v}_t) = \frac{1}{1-t} \left[ p_\theta(\boldsymbol{E}^{i,j}|\mathbf{v}_t, \boldsymbol{E}_t) - \delta(\boldsymbol{E}^{i,j}_t) \right].$$

The denoiser model $p_\theta(G_1|G_t)$ outputs probabilities of a "clean" (reactant) graph over nodes and edges conditioned on a noisy state graph $G_t$, an interpolation between synthons and reactant graphs. The stochastic processes over nodes and edges are coupled due to the denoiser model taking the noisy graph as input. We can express the outputs of the denoiser model separately over the nodes and edges $p_\theta(\mathbf{v}|\mathbf{v}_t, \boldsymbol{E}_t)$ and $p_\theta(\boldsymbol{E}|\mathbf{v}_t, \boldsymbol{E}_t)$. This leads to a natural objective for discrete flow matching where we minimize the weighted combination of the cross-entropy loss over nodes and edges:

$$\mathcal{L}(\theta) = -\mathbb{E}_{p(t),p_0(G_0),p_1(G_1),p_t(G_t|G_0,G_1)} \left[ \sum_i \log p_\theta(\mathbf{v}^i|\mathbf{v}_t, \boldsymbol{E}_t) + \lambda \sum_{i,j} \log p_\theta(\boldsymbol{E}^{i,j}|\mathbf{v}_t, \boldsymbol{E}_t) \right],$$

where the distribution over time $p(t)$ is sampled uniformly, i.e. $t \sim \mathcal{U}(0, 1)$.

As in the previous case, we append "dummy" nodes (atoms) to the synthon molecule graph since the reactant molecule graph can have atoms that are not present in the synthon graph. For every synthon graph $G^s_i$, we generate $N_i$ sets of reactants. The $N_i$'s and $M$ are hyperparameters. Given a budget of $N$ set of reactants per product, we constrain $\sum_{i=1}^M N_i = N$. We demonstrate in our experiment section that these hyperparameters do not require much tuning, and $M = 2$ provides SOTA performance.

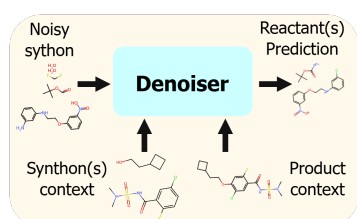

Figure 3: Overview of flow matching denoiser $p_\theta$.

We adopt the reaction center prediction model from Shi et al. [39] to identify synthons. We do note however, that any model that outputs synthons could be equivalently used. Prior works such as [54, 39, 41] typically treat reaction center identification as a bond-level classification task. They use graph neural networks to classify each bond in the product as reactive or not by predicting bond-level reactivity scores or edit scores. During inference, we select the top-$M$ bonds above a certain score threshold as candidate reaction centers, yielding $M$ synthon predictions.

## 3.2 Reward-based steering

Given the discrete flow matching framework for retrosynthesis above, we can specify a potential function $U_t$ and reward $r(\mathbf{x}_1)$ to perform inference time steering. For intermediate steps $t < 1$, we define:

$$U_t(\mathbf{x}_{0:t}) = \exp\left(\sum_{s=0}^t r_\phi(\mathbf{x}_s)\right), \quad \text{and} \quad U_1 = \exp(\lambda r_\phi(\mathbf{x}_1)) \left(\prod_{t \in \{0, \cdots, 1-h\}} U_t\right)^{-1}.$$

This design ensures that $p_{\text{FK},1} \propto p_{\text{target}}$ while steering intermediate particles toward high-reward trajectories. Furthermore, it selects particles that have the highest accumulated reward.

To perform SMC resampling, we need access to a reward oracle, $r_\phi$ that models the distribution of rewards $p_\theta(r(\mathbf{x}_1)|\mathbf{x}_t)$ generated from the intermediate state $\mathbf{x}_t$. Fortunately, we can still obtain high-quality estimates from this reward distribution without training a separate reward model by querying the flow matching denoiser model $p_\theta(\mathbf{x}_1|\mathbf{x}_t)$ instead. Specifically, the intermediate reward is defined as $r_\phi(\mathbf{x}_t) := r(\hat{\mathbf{x}}_1)$ where $\hat{\mathbf{x}}_1 = \mathbb{E}_{p_\theta(\mathbf{x}_1|\mathbf{x}_t)}[\mathbf{x}_1|\mathbf{x}_t]$ is the expected $\mathbf{x}_1$ given $\mathbf{x}_t$. This choice of intermediate reward can be evaluated efficiently without significant additional computational cost.

Our reward function $r_\phi(G_1)$ is inspired by round-trip accuracy, which measures the ability of a retrosynthesis model to recover diverse and feasible reactants. There may be many different sets of reactants that can synthesize the same product. Top-$k$ round-trip accuracy aims to capture this characteristic of retrosynthesis by quantifying the proportion of feasible reactants(s) among the top-$k$ predictions from the retrosynthesis model. We can assess whether a reaction is feasible by using a forward-synthesis model, which predicts the product molecule produced by a set of reactants. A reaction is deemed feasible if the forward model satisfies $F(\hat{G}^r) = G^p$, where $\hat{G}^r$ is the predicted reactants. The reward function serves as a proxy for round-trip accuracy, encouraging the generation of chemically valid and synthetically feasible reactants. Suppose $(G^p, G^r)$ is a pair of product and reactant molecule graphs. The reward for the intermediate state $G_t$ for the flow matching process on graphs is defined as $r_\phi(G_t) = \mathbb{1}[F(\hat{G}_1) = G^p]$ where $\hat{G}_1$ is the expected $G_1$ (reactants) given $G_t$. With this formulation of the reward, we introduce RETRO PRODFLOW-RS, a reward-steered version for RETRO PRODFLOW. Finally, we use Molecular Transformer [35] for the forward-synthesis model.

## 4 Experiments and Results

We evaluate our proposed methods against state-of-the-art template-free and template-based models on standard single-step retrosynthesis benchmarks. Through RETRO SYNFLOW, we aim to highlight the effectiveness of synthons as an inductive prior for generating reactants using flow matching. Additionally, we evaluate RETRO PRODFLOW-RS to show that inference-time reward-based steering enhances the diversity and feasibility of predicted reactants. We conduct various ablation studies assessing the performance of FK-steering and SMC-based resampling on standard metrics. Unless otherwise stated, we generate $N = 100$ sets of reactants for each input product. Our methods discretize the time interval $[0, 1]$ into $T = 50$ steps. For SMC resampling, RPF-RS uses $K = 4$ particles. For RSF, we use $M = 2$ synthon predictions with $N_1 = 70$ and $N_2 = 30$ for the top-2 synthon predictions respectively. Code is available at: https://github.com/DSL-Lab/RetroSynFlow.

### 4.1 Experimental Setup

**Dataset**. We trained and evaluated our methods on the USPTO-50K dataset [34], a standard benchmark for retrosynthesis modelling containing 50k atom-mapped reactions extracted from US patents. We follow the same train/evaluation/test split used by RetroBridge [18] and GLN [9]. As done in Retrobridge, we randomly permute the graph nodes as a pre-processing step before input to the flow matching model.

**Baselines**. We evaluate our methods against both template and template-free baselines. On the template-free side, we compare against graph-based approaches such as RetroBridge [18] and G2G [39]. We compare against SMILE string translation approaches such as Tied-Transformer [20], Augmented-Transformer [48], and SCROP [57]. Our baselines also include methods that combine graph and SMILE representations: GTA [38], MEGAN [33], Dual-TF [46], and Graph2SMILES [49]. On the template-based side, we compare against state-of-the-art approaches such as GLN [9], GraphRetro [41], LocalRetro [4], and RetroGFN [13]. We also compare against Chimera [27], a framework that ensembles multiple different SMILE and graph based models across template and template-free approaches.

**Evaluation**. We report top-$k$ exact match accuracy, which measures the proportion of reactions where the ground-truth set of reactants is in the top $k$ set of reactants predicted by the model. Following standard practices in prior works, we report $k = 1, 3, 5, 10$. Additionally, we report top-$k$ round-trip accuracy and round-trip coverage to measure reaction feasibility. Given product and reactant molecular graphs $(G^p, G^r)$, let $\mathcal{R} = \{\hat{G}_i^r\}_{i=1}^k$ be the top-$k$ predicted sets of reactants for $G^p$. Let $\mathcal{P} = \{F(\hat{G}_i^r)\}_{i=1}^k$ be the predicted products from the forward-synthesis model. Then

top-$k$ exact-match accuracy, round-trip accuracy and coverage for the product $\mathbf{x}$ are computed as follows: $\mathbb{1}[G^r \in \mathcal{R}]$, $\frac{1}{k}\sum_{i=1}^{k} \mathbb{1}[G^p = F(\hat{G}_i^r)]$, and $\mathbb{1}[G^p \in \mathcal{P}]$.

## 4.2 Main Results

Table 1: Top-$k$ accuracy (exact match) on the USPTO-50k test dataset.

| | Model | Top-$k$ Accuracy | | | |
| | | $k=1$ | $k=3$ | $k=5$ | $k=10$ |
|---|---|---|---|---|---|
| **TB** | GLN | 52.5 | 74.7 | 81.2 | 87.9 |
| | GraphRetro | **53.7** | 68.3 | 72.2 | 75.5 |
| | LocalRetro | 52.6 | **76.0** | **84.4** | **90.6** |
| | RetroGFN | 49.2 | 73.3 | 81.1 | 88.0 |
| **TF** | SCROP | 43.7 | 60.0 | 65.2 | 68.7 |
| | G2G | 48.9 | 67.6 | 72.5 | 75.5 |
| | Aug. Transformer | 48.3 | — | 73.4 | 77.4 |
| | DualTF$_{aug}$ | 53.6 | 70.7 | 74.6 | 77.0 |
| | MEGAN | 48.0 | 70.9 | 78.1 | 85.4 |
| | Tied Transformer | 47.1 | 67.1 | 73.1 | 76.3 |
| | GTA$_{aug}$ | 51.1 | 67.0 | 74.8 | 81.6 |
| | Graph2SMILES | 52.9 | 68.5 | 70.0 | 75.2 |
| | Retroformer$_{aug}$ | 52.9 | 68.2 | 72.5 | 76.4 |
| | Chimera [1] | 59.6 | **82.8** | **89.2** | **94.2** |
| | RetroBridge | 50.8 | 74.1 | 80.6 | 85.6 |
| | RETRO PRODFLOW | $50.0 \pm 0.15$ | $74.3 \pm 0.30$ | $81.2 \pm 0.08$ | $85.8 \pm 0.04$ |
| | RETRO SYNFLOW | $\mathbf{60.0 \pm 0.22}$ | $77.9 \pm 0.13$ | $82.7 \pm 0.15$ | $85.3 \pm 0.19$ |

Our main results are presented in Table 1, comparing RETRO SYNFLOW to several previous SOTA models for retrosynthesis on top-$k$ accuracy, and Table 2 compares RETRO PRODFLOW-RS on top-$k$ round-trip accuracy. In particular, RETRO PRODFLOW-RS achieves a top-1 accuracy of $60\%$, approximately 20 % higher than other template-free methods that do not perform ensembling. It outperforms RetroBridge and RETRO PRODFLOW, which use a source distribution of product molecules to build the Markov Bridge/CTMC [2]. This demonstrates that synthons encode valuable structural information and serve as a more informative source of distribution for generating reactants. Additionally, RETRO SYNFLOW also produces notable gains in top-3 and top-5 exact match accuracy. Our approach is competitive with Chimera [27], a framework that ensembles many different retrosynthesis models across both graph and SMILE string-based methods. As a result, our method is complementary to their framework and may be less resource intensive.

Table 2: Top-$k$ Round-trip coverage and accuracy on USPTO-50k test dataset.

| | Model | Round-Trip Coverage | | | | Round-Trip Accuracy | | | |
| | | $k=1$ | $k=3$ | $k=5$ | $k=10$ | $k=1$ | $k=3$ | $k=5$ | $k=10$ |
|---|---|---|---|---|---|---|---|---|---|
| **TB** | GLN | **82.5** | 92.0 | 94.0 | – | **82.5** | **71.0** | 66.2 | – |
| | LocalRetro | 82.1 | **92.3** | **94.7** | – | 82.1 | **71.0** | **66.7** | – |
| | RetroGFN | – | – | – | – | 76.7 | 69.1 | 65.5 | 60.8 |
| **TF** | MEGAN | 78.1 | 88.6 | 91.3 | – | 78.1 | 67.3 | 61.7 | – |
| | Graph2SMILES | – | – | – | – | 76.7 | 56.0 | 46.4 | – |
| | Retroformer$_{aug}$ | – | – | – | – | 78.6 | 71.8 | 67.1 | – |
| | RetroBridge | 85.1 | 95.7 | 97.1 | 97.7 | 85.1 | 73.6 | 67.8 | 56.3 |
| | RETRO SYNFLOW-RS | 88.9 | 97.3 | 98.4 | 98.9 | 88.9 | 73.9 | 69.1 | 61.8 |
| | RETRO PRODFLOW-RS | **91.4** | **97.6** | **98.7** | **99.3** | **91.4** | **84.1** | **80.1** | **75.1** |

---

[1]Chimera [27] an ensemble method combining multiple retrosynthesis models (TF + TB)

[2]RetroBridge was evaluated using $T = 500$ sampling steps as done in Igashov et al. [18]

However, exact match accuracy is limited because it does not capture the fact that multiple reactants (some of which may not exist in the dataset) can synthesize the same product molecule. Therefore, we evaluate RETRO PRODFLOW-RS on round-trip accuracy and compare it to the baselines present in [18] in addition to [13]. As shown in Table 2, RETRO PRODFLOW-RS is capable of generating diverse and feasible reactants, outperforming state-of-the-art methods on round-trip coverage and accuracy. Additionally, RETRO SYNFLOW-RS also achieves competitive results on round-trip coverage and accuracy.

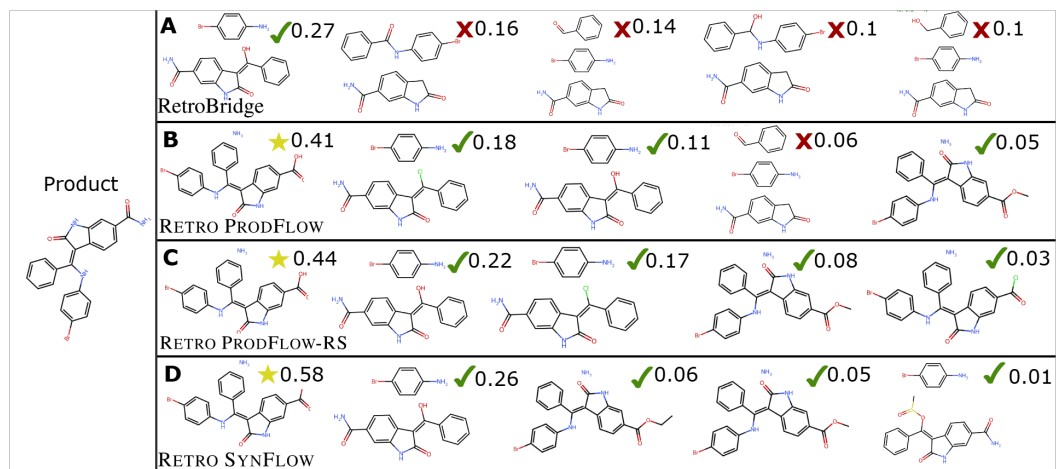

Figure 4: Top-5 reactants selected by each method. A star indicates an exact match, a checkmark indicates a round-trip match but not an exact match, and a cross means neither.

## 4.3 Ablation Studies

In Table 3, we show the performance of RETRO PRODFLOW-RS for the synthons to reactants generation task. Here, top-$k$ accuracy refers to the proportion of synthons where the true set of reactants exists in the top-$k$ predicted sets of reactants. Furthermore, we evaluate the effects of providing the product molecule as additional context to the flow matching model $p_\theta$.

Table 3: Top-$k$ accuracy for synthon completion task on USPTO-50k test dataset.

| Model | 1 | 3 | 5 | 10 |
|---|---|---|---|---|
| RSF (w/o product) | 59.7 | 75.5 | 79.1 | 82.0 |
| RSF (w product) | **67.7** | **82.9** | **85.7** | **87.5** |

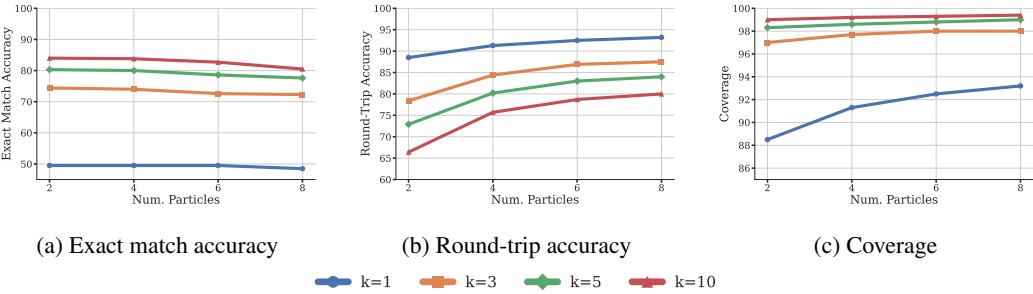

(a) Exact match accuracy     (b) Round-trip accuracy     (c) Coverage

k=1    k=3    k=5    k=10

Figure 5: Performance of RETRO PRODFLOW-RS on the USPTO-50k validation set as we vary the number of particles for SMC resampling. We sample $N = 50$ reactants per product.

In Figure 5, we study the performance of RETRO PRODFLOW-RS by varying the number of particles in the SMC resampling procedure for FK-steering. We find that $K = 4$ particles already provides significant gains in round-trip accuracy and coverage with negligible reduction in exact match

accuracy. As we increase the number of particles further, we see noticeable increases in round-trip accuracy and coverage with a slight decrease in exact match accuracy.

Table 4: Comparing RETRO PRODFLOW-RS against baselines on the USPTO-50k test set.

| Model | Round-Trip Coverage | | | | Round-Trip Accuracy | | | |
|---|---|---|---|---|---|---|---|---|
| | $k=1$ | $k=3$ | $k=5$ | $k=10$ | $k=1$ | $k=3$ | $k=5$ | $k=10$ |
| RPF | 84.4 | 95.3 | 96.9 | 97.7 | 84.4 | 72.8 | 66.8 | 57.6 |
| RPF-RS | **91.4** | **97.6** | **98.7** | **99.3** | **91.4** | **84.1** | **80.1** | **75.1** |
| Greedy Sampling | 89.4 | 97.0 | 98.4 | 99.2 | 89.4 | 79.0 | 73.0 | 66.2 |
| RPF (400 reactants) | 84.2 | 95.3 | 97.1 | 98.1 | 84.2 | 73.1 | 68.5 | 60.7 |

Next, we further demonstrate the benefits of reward-based steering with SMC resampling by comparing RETRO PRODFLOW-RS against two baselines. The greedy sampling approach does not perform any steering or resampling and instead selects the particle with the highest reward at the end of the generation process. Table 4 shows that RETRO PRODFLOW-RS outperforms greedy sampling, highlighting the superiority of SMC resampling. Furthermore, we also scale the computational budget of RETRO PRODFLOW by generating $N = 400$ reactants per product molecule. This has the same computational cost as RETRO PRODFLOW-RS, which uses $K = 4$ particles and samples 100 reactants per product molecule. Again, Table 4 shows that increasing the number of reactants sampled per product does not improve round-trip accuracy and coverage by a significant amount, highlighting the need for reward-based SMC steering. Additional ablation studies are available in Appendix B.

## 5   Related Works

**Template-based**. The approaches for single-step retrosynthesis can be divided into two main categories: template-based and template-free. Reaction templates are molecular subgraph patterns that encode pre-defined reaction rules to transform a target product into simpler reactants. They can be hand-crafted by experts [15, 47] or extracted algorithmically from large databases [5]. The main challenge for template-based methods, such as Segler and Waller [36], Coley et al. [5], Dai et al. [9], is ranking and selecting the correct templates for a target molecule. More recently, Gaiński et al. [13] utilizes the recent GFlowNet framework to build RetroGFN, a model capable of composing existing reaction templates to explore the solution space of reactants beyond the dataset to increase feasibility and diversity.

**Template-free**. Templates provide a strong inductive bias, and template-based methods offer greater interpretability at the expense of generalization. On the other hand, template-free approaches directly transform products to reactants without pre-defined rules, providing more flexibility. Many works in this area frame the task as a sequence-to-sequence modelling problem on SMILES string representation of molecules [24, 57, 46, 48]. Another approach is to use a graph representation of molecules and transform product molecule graphs to reactant graphs [33, 39]. The recent work by Igashov et al. [18] builds a Markov Bridge model between the space of products and reactants. Also, Laabid et al. [22] employs absorbing state diffusion and builds a graph diffusion model to generate reactants. Other works leverage a combination of graph-based and SMILE representations of molecules [38, 49, 52]. In the context of multi-step retrosynthesis, prior works have used a forward-synthesis model to select for promising reactants [7, 37, 57].

**Discrete Diffusion and FM**. Discrete diffusion and flow matching are a powerful class of generative models that have demonstrated impressive results across various tasks, *e.g.* language modelling [28], symmetric group learning [56], and biological applications such as protein synthesis[1, 17], 3D molecule generation [10, 42], and DNA sequence design [44]. Traditionally, continuous-state diffusion models have also been employed for discrete data generation tasks such as graph synthesis [55, 53, 19, 29], despite relying on hard-coded post-processing steps.

## 6   Conclusion

In this work, we approached single-step retrosynthesis as learning the transport map between two intractable distributions and explored two different options for the source. We first introduced RETRO

PRODFLOW, a discrete flow matching model that transforms products into reactants. Next, we proposed RETRO SYNFLOW, the first flow matching model that transforms products to reactants through intermediary structures known as synthons. We demonstrated that synthons serve as a more informative source of distribution for generating reactants with RETRO SYNFLOW achieving 60% top-1 accuracy, beating previous SOTA methods for retrosynthesis modelling. Furthermore, we enhanced the diversity and feasibility of predictions by leveraging Feynman-Kac steering, an inference-time, reward-based steering method. We define the reward function using a forward-synthesis model motivated by round-trip accuracy. A reward-steered version of RETRO PRODFLOW achieves 80.1% top-5 round-trip accuracy, beating template-free SOTA methods by up to 19%.

Although template-based planning is constrained, it may remain preferable to many chemists due to its alignment with established reactants, reagents, and reaction types. Future work could explore incorporating template information to guide the generation process toward specific compound sets, enhancing the interpretability of our method.

## 7    Acknowledgements

This work was funded, in part, by the NSERC DG Grant (No. RGPIN-2022-04636), the Vector Institute for AI, Canada CIFAR AI Chairs, a Google Gift Fund, and the CIFAR Pan-Canadian AI Strategy through a Catalyst award. Resources used in preparing this research were provided, in part, by the Province of Ontario, the Government of Canada through the Digital Research Alliance of Canada alliance.can.ca, and companies sponsoring the Vector Institute www.vectorinstitute.ai/#partners, and Advanced Research Computing at the University of British Columbia. Additional hardware support was provided by John R. Evans Leaders Fund CFI grant. AJB is partially supported by an NSERC Postdoc fellowship and supported by the EPSRC Turing AI World-Leading Research Fellowship No. EP/X040062/1 and EPSRC AI Hub No. EP/Y028872/1. QY is supported by UBC Four Year Doctoral Fellowships.

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

# Supplementary Materials

## Table of Contents

## A  Experimental details

In this section, we elaborate on our experimental setup. We view a set of molecules as a single potentially disconnected graph. We follow the procedure outlined in Igashov et al. [18] and introduce a "dummy" node as an atom type. The graph of reactant molecules has at least as many nodes as the product molecule graph. During training and inference for RETRO PRODFLOW, we append ten dummy nodes to each product molecule. This covers $99.4\%$ of the reactions in the USPTO-50k test dataset. Following Igashov et al. [18], the remaining reactions are removed from the test data. During inference, these dummy nodes are potentially transformed into true atom nodes. Similarly, we also append ten dummy nodes to the synthon molecule graphs when using RETRO SYNFLOW. There are 16 atom types (not including dummy atoms) and 4 bond types (not including no bond). Our methods are implemented in PyTorch [30], and we also use an open-source software RDKit [23], for operations involving chemical reactions and molecular graphs.

### A.1  Neural Network Model

We use a graph transformer network [11, 51] also used by Igashov et al. [18] to model the denoiser $p_\theta$ of the flow matching process. The denoiser model takes a noisy graph $(\mathbf{v}, \boldsymbol{E})$ and graph-level features $\mathbf{y}$ as input and outputs probabilities of graphs over the data distribution. In the case of RETRO PRODFLOW, the product molecule graph is also provided as input to the denoiser model. This is done by appending the product molecule graph's node feature vector and adjacency matrix to the node feature vector and adjacency matrix of the noisy graph. For RETRO SYNFLOW, both the product molecule graph and synthon molecule graph are provided as input.

The graph transformer network is similar to the standard transformer architecture and consists of a graph attention module depicted in Figure 6. The graph attention module takes in input node features $\mathbf{v}$, edge features $\boldsymbol{E}$, and graph-level features $\mathbf{y}$. The FiLM is defined as $\text{FiLM}(\boldsymbol{M}_1, \boldsymbol{M}_2) = \boldsymbol{M}_1 \boldsymbol{W}_1 + (\boldsymbol{M}_1 \boldsymbol{W}_2) \odot \boldsymbol{M}_2 + \boldsymbol{M}_2$ where $\boldsymbol{W}_1, \boldsymbol{W}_2$ are learnable weights. Also,

PNA is defined as $\text{PNA}(\mathbf{v}) = \text{cat}(\max(\mathbf{v}), \min(\mathbf{v}), \text{mean}(\mathbf{v}), \text{std}(\mathbf{v}))\boldsymbol{W}$ where $\boldsymbol{W}$ is a learnable weight.

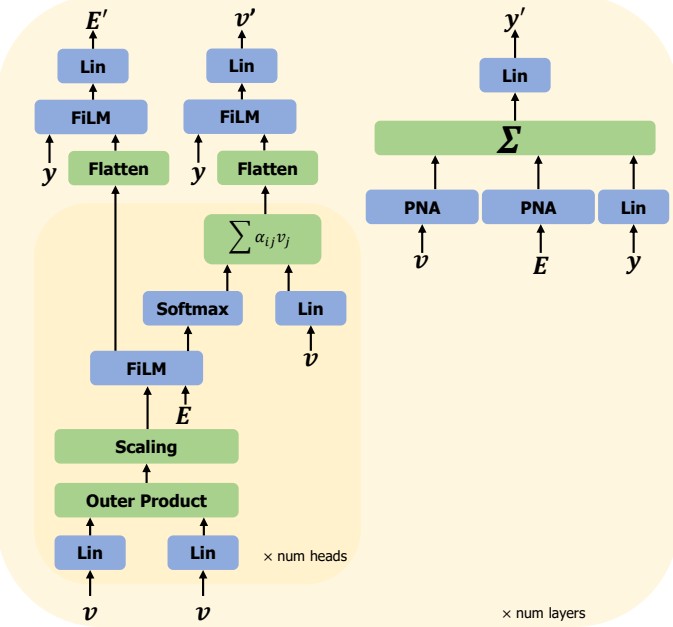

Figure 6: An overview of the graph attention module used in the graph transformer network. The output features are passed through a normalization layer and a fully connected layer at the end.

## A.2 Training

Our training runs are done on either an NVIDIA RTX 3090 (24 GB of memory) or V100 (32 GB of memory). We train all of our models up to 600 epochs which can take up to 32 hours. We compute top-$k$ accuracy metrics on a portion of the validation set every fixed number of epochs and select the checkpoint that has the highest top-1 accuracy. The models are trained using a batch size of 32. We use AdamW [26] with a learning rate of 0.0002.

## A.3 Additional Features

We utilize the additional features proposed by [51] and used in Igashov et al. [18] as input to our models. We briefly state these features here for completeness.

**Cycles**. Message Passing Neural Networks cannot detect graph cycles, so we add them as features using formulas up to cycles of size 6. We compute node-level features (how many cycles does this node belong to) up to size 5 and graph-level features (how many cycles does this graph have) up to size 6. Fortunately, we can use formulas to compute the graph-level features $\mathbf{y}_i$ and node-level features $\mathbf{X}_i$, which can be efficiently computed on the GPU. In the following formulas, $d$ denotes the

vector containing node degrees and $\|\cdot\|_F$ denotes the Frobenius norm:

$$\mathbf{X}_3 = \text{diag}(\mathbf{A}^3)/2$$

$$\mathbf{X}_4 = \left(\text{diag}(\mathbf{A}^4) - d(d-1) - \mathbf{A}(d\mathbf{1}_n^T)\mathbf{1}_n\right)/2$$

$$\mathbf{X}_5 = \left(\text{diag}(\mathbf{A}^5) - 2\,\text{diag}(\mathbf{A}^3)\odot d - \mathbf{A}((\text{diag}(\mathbf{A}^3)\mathbf{1}_n^T)\mathbf{1}_n) + \text{diag}(\mathbf{A}^3)\right)/2$$

$$\mathbf{y}_3 = \mathbf{X}_3^T\mathbf{1}_n/3$$

$$\mathbf{y}_4 = \mathbf{X}_4^T\mathbf{1}_n/4$$

$$\mathbf{y}_5 = \mathbf{X}_5^T\mathbf{1}_n/5$$

$$\mathbf{y}_6 = \text{Tr}(\mathbf{A}^6) - 3\,\text{Tr}(\mathbf{A}^3 \odot \mathbf{A}^3) + 9\|\mathbf{A}(\mathbf{A}^2 \odot \mathbf{A}^2)\|_F$$
$$- 6\left\langle\text{diag}(\mathbf{A}^2), \text{diag}(\mathbf{A}^4)\right\rangle + 6\,\text{Tr}(\mathbf{A}^4) - 4\,\text{Tr}(\mathbf{A}^3)$$
$$+ 4\,\text{Tr}(\mathbf{A}^2\mathbf{A}^2 \odot \mathbf{A}^2) + 3\|\mathbf{A}^3\|_F - 12\,\text{Tr}(\mathbf{A}^2 \odot \mathbf{A}^2) + 4\,\text{Tr}(\mathbf{A}^2).$$

**Spectral Features**. We compute graph-level features: the number of connected components (which is the multiplicity of the 0 eigenvalue), and the first 5 non-zero eigenvalues of the graph Laplacian. We also compute node-level features: an estimate of the biggest connected component and the first two eigenvectors associated with the first two non-zero eigenvalues. Since molecular graphs in USPTO-50k have fewer than 100 nodes, the computation of these spectral features is not a concern.

# B Additional Ablation Studies

## B.1 Synthon Prediction

In this section, we provide some additional ablation studies examining the performance of our methods. In Table 5, we evaluate the performance of RETRO PRODFLOW-RS when sampling $N = 100$ reactants with $M = 2$ synthon predictions. We vary $N_1$, the number of reactant predictions generated for the highest ranking synthon prediction from the reaction center identification model. We verify that we need to generate more reactants for the highest-scoring synthon prediction to obtain competitive top-$k$ accuracy.

|       | **Top-$k$ Accuracy** | | | |
|-------|-------|-------|-------|--------|
| $N_1$ | $k=1$ | $k=3$ | $k=5$ | $k=10$ |
| 90    | 58.2  | 77.4  | 81.9  | 84.4   |
| 80    | 58.3  | 78.0  | 82.3  | 84.7   |
| 70    | 58.1  | 77.5  | 82.0  | 84.6   |
| 60    | 56.6  | 77.0  | 81.6  | 84.5   |
| 50    | 48.5  | 76.1  | 81.2  | 84.2   |

Table 5: Top-$k$ accuracy (exact match) on the USPTO-50k validation dataset of RETRO SYNFLOW with $M = 2$ synthon predictions, sampling $N = 100$ reactants and varying the split. Given $N_1$, we have $N_2 = 100 - N_1$.

## B.2 Sampling Steps

Next, we conduct a study to understand how the number of sampling steps affects the performance of flow matching compared to RetroBridge. We find that $T = 50$ sampling steps is sufficient for RETRO PRODFLOW to obtain SOTA results. Although RetroBridge achieves a higher accuracy at $T = 5$ or $T = 10$ sampling steps compared to flow matching, both methods fail to reach competitive performance. At $T = 50$ steps, RETRO PRODFLOW achieves some improvement over RetroBridge.

We also investigate the inference time required by RETRO SYNFLOW to sample $N = 100$ reactants with $T = 50$ sampling steps. We run this test on an RTX 3090 with 24 GB of memory. The average time required to sample 100 reactants for each product molecule in USPTO-50k test dataset is $5.46 \pm 2.95$ seconds.

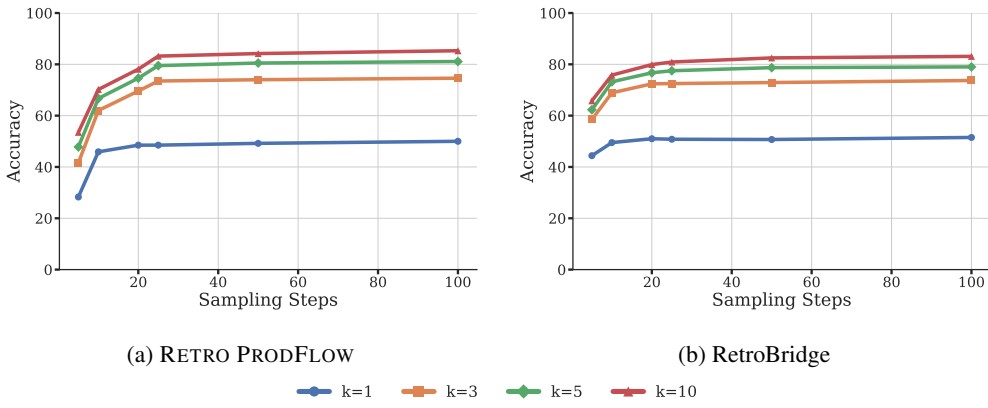

(a) RETRO PRODFLOW  (b) RetroBridge

k=1  k=3  k=5  k=10

Figure 7: The performance of RETRO PRODFLOW and RetroBridge as we vary the number of sampling steps.

## B.3 Inference Time Comparison

In this section, we provide an ablation comparing the inference time scaling of RSF with the number of particles against other template-based and template-free models. Our model has 3.38 million parameters. We benchmark on an RTX 3090 with 24 GB of memory.

| Method | Mean time |
|---|---|
| GLN | 0.295 |
| LocalRetro | 0.022 |
| RetroBridge | 56.3 |
| RSF (K=1) | 5.4 |
| RSF (K=2) | 15.6 |
| RSF (K=4) | 30.6 |
| RSF (K=6) | 45.7 |
| RSF (K=8) | 53.9 |

Table 6: Average inference time (seconds) to sample 100 reactants for a given product in the USPTO-50k test set.

We note that 30 seconds for sampling a set of 100 reactants for a given product is a completely feasible time for applications of models like ours. In particular, this speed does not prevent the use of our method as a component of a multistep retrosynthesis planning pipeline. Additionally, the sampling time for RetroBridge with T=500 steps and 100 reactants is 50 seconds.

## C  Round-trip Visualization

This section provides a short case study analyzing the outputs of RETRO PRODFLOW and RETRO PRODFLOW-RS with $K = 2$ particles. We aim to understand how top-$k$ accuracy can decrease when applying inference-time steering to guide generations towards outputs that optimize round-trip accuracy. We look at the top-1 accuracy results on the USPTO-50k test dataset for simplicity. Our main finding from this ablation is that it is still possible for RETRO PRODFLOW to generate reactants that are incorrect, *i.e.*, do not match the true reactants and are not feasible, *i.e.*, the forward synthesis model prediction does not match the ground-truth product. Table 7 shows how steering based on a round-trip reward affects the incorrect/correct prediction made by RETRO PRODFLOW. In total, 257 correct examples in the test dataset get converted to incorrect examples when applying reward steering. On the other hand, 251 incorrect examples are converted to correct examples when applying steering. As we increase the number of particles, *i.e.*, increase the strength of steering, this gap widens. This results in an overall decrease in exact-match accuracy as we force reactants towards

more diverse and feasible predictions. Figures 8 and 9 show the visualizations between the outputs of RETRO PRODFLOW and RETRO PRODFLOW-RS. The predicted product column refers to the prediction of the forward-synthesis model given the predicted reactants as input.

Table 7: Top-1 predicted reactants from RPF and RPF-RS quantified into four categories. The round-trip match column indicates whether the prediction made by RPF-RS is a round-trip match.

| RPF | RPF-RS | Round-Trip Match | Count | Percentage |
|---|---|---|---|---|
| Correct | Incorrect | T | 225 | 4.5 |
| Correct | Incorrect | F | 32 | 0.64 |
| Incorrect | Correct | T | 226 | 4.5 |
| Incorrect | Correct | F | 25 | 0.50 |
| Correct | Correct | T | 1848 | 36.9 |
| Correct | Correct | F | 388 | 7.7 |
| Incorrect | Incorrect | T | 1810 | 36.1 |
| Incorrect | Incorrect | F | 453 | 9.0 |

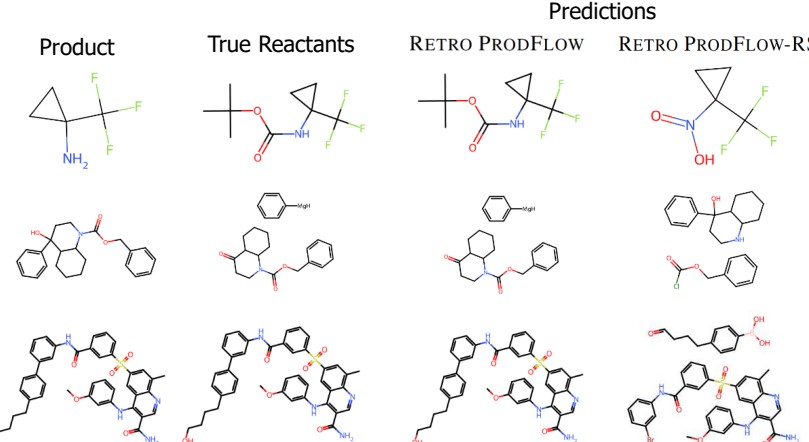

Figure 8: A visualization of reactions where RETRO PRODFLOW-RS generates an incorrect reactant prediction that is still feasible, while the RETRO PRODFLOW generates the correct reactant prediction. There are 225 examples in the test set that correspond to this case.

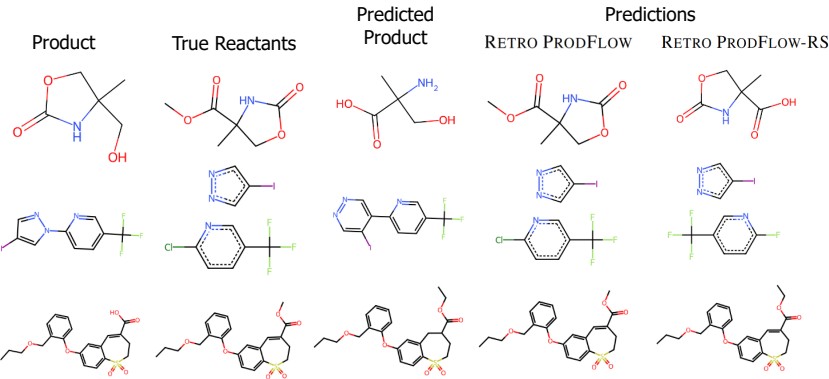

Figure 9: A visualization of reactions where RETRO PRODFLOW-RS generates an incorrect reactant prediction that is infeasible. RETRO PRODFLOW generates the correct reactant prediction. There are 32 examples in the test set that correspond to this case.

To obtain a better understanding of the errors of the forward-synthesis model, we evaluate the top-$k$ accuracy of Molecular Transformer on the USPTO-50K dataset. Furthermore, we find that for the

non-steering outputs, 25% of the chemically valid reactants were misscored by the forward-synthesis model. For the FK-steered outputs, only 14% of the chemically valid reactants were misscored by the forward-synthesis model.

| Top-1 | Top-3 | Top-5 | Top-10 |
|-------|-------|-------|--------|
| 75.0 | 82.0 | 83.0 | 83.0 |

Table 8: Top-$k$ accuracy of Molecular Transformer on the USPTO-50k evaluation set.

## D  Additional Sampling Scheme

Recently, there has been increasing interest in developing advanced adaptive sampling schemes for discrete diffusion and flow matching models [16, 32, 21, 31]. These developments aim to reduce errors in the generation process while improving inference speed. As explained in Section 2.1, we update the intermediate sample $\mathbf{x}_t$ using the following transition kernel, $\mathbf{x}_{t+h}^i \sim \text{Cat}(\mathbf{x}_{t+h}^i; \delta(\mathbf{x}_t^i) + hu_t^i(\mathbf{x}_{t+h}^i, \mathbf{x}_t))$, which is analogous to the Euler update step in continuous flow matching. Inspired by this analogy, we explore a higher-order sampling scheme [32] based on the Runge-Kutta (RK) method for solving ODEs. The update step for the method is as follows:

$$\hat{\mathbf{x}}_{t+h}^i \sim \text{Cat}(\hat{\mathbf{x}}_{t+h}^i; \delta(\mathbf{x}_t^i) + hu_t^i(\hat{\mathbf{x}}_{t+h}^i, \mathbf{x}_t)) \tag{3}$$

$$\mathbf{x}_{t+h}^i \sim \text{Cat}\left(\mathbf{x}_{t+h}^i; \delta(\mathbf{x}_t^i) + \frac{1}{2}hu_t^i(\mathbf{x}_{t+h}^i, \mathbf{x}_t) + \frac{1}{2}hu_{t+h}^i(\mathbf{x}_{t+h}^i, \hat{\mathbf{x}}_{t+h}^i)\right). \tag{4}$$

This update step requires two model evaluations from $p_\theta$ instead of one. Table 9 compares the performance of RETRO PRODFLOW using this sampling scheme with 25 steps against RETRO PRODFLOW using the Euler-inspired sampling scheme with 50 steps.

Table 9: Top-$k$ accuracy of RPF on the USPTO-50k test set sampling $N = 50$ reactants per product.

| Model | 1 | 3 | 5 | 10 |
|-------|-----|-----|-----|-----|
| RPF | 49.6 | 73.3 | 79.6 | 83.6 |
| RPF (RK 25 steps) | 49.3 | 71.2 | 76.4 | 80.0 |
| RPF (RK 50 steps) | 49.3 | 72.5 | 78.6 | 82.3 |

## E  Predictions Visualization

We provide some additional visualizations of the generated reactants from our methods. In the following figures, an "E" represents an exact-match between the prediction reactant and an "R" indicates a round-trip match but not an exact-match. We show the top-3 reactants.

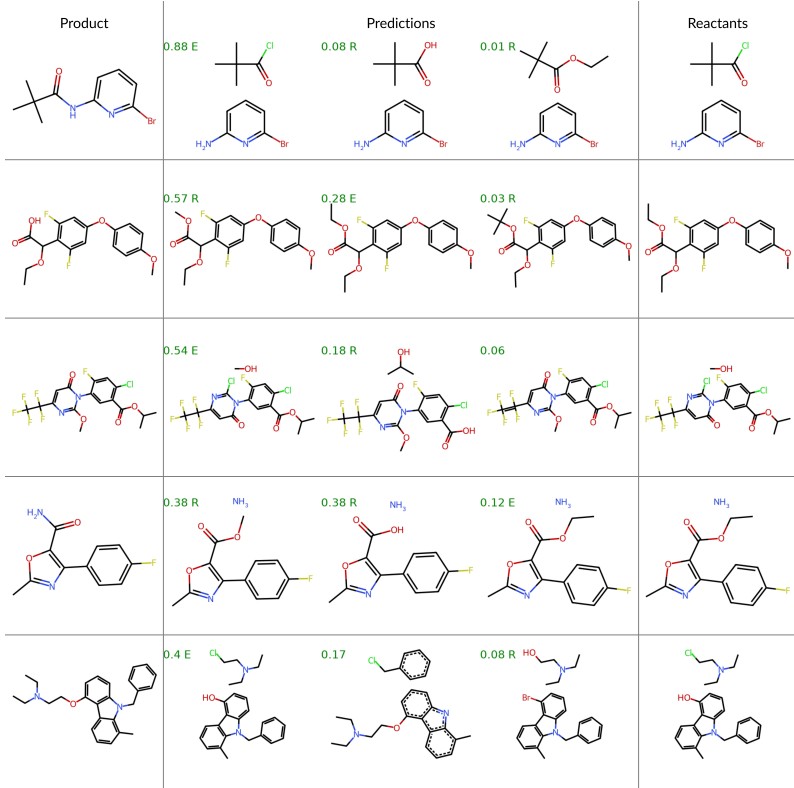

Figure 10: Visualizations of predictions made by RETRO PRODFLOW. Examples are taken from the USPTO-50k test set randomly.

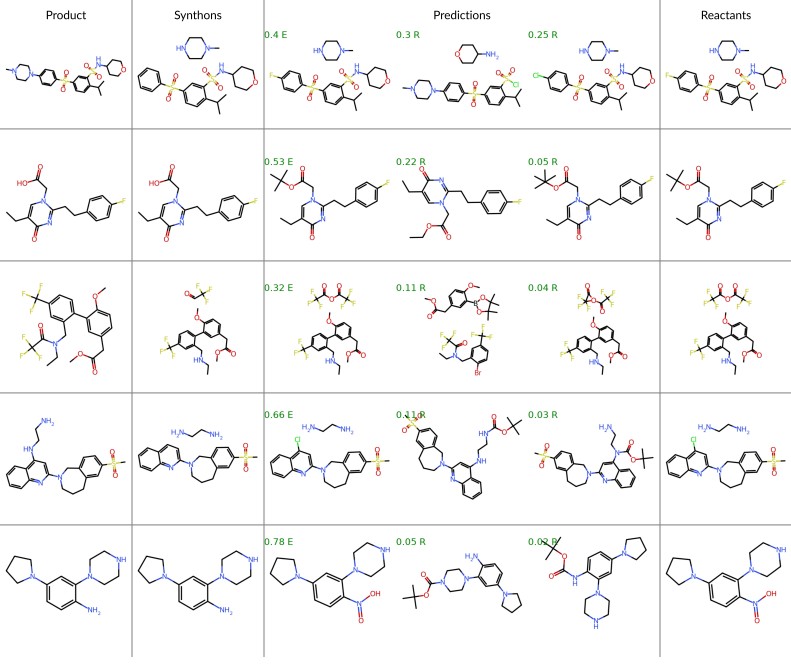

Figure 11: Visualizations of predictions made by RETRO SYNFLOW. Examples are taken from the USPTO-50k test set randomly.

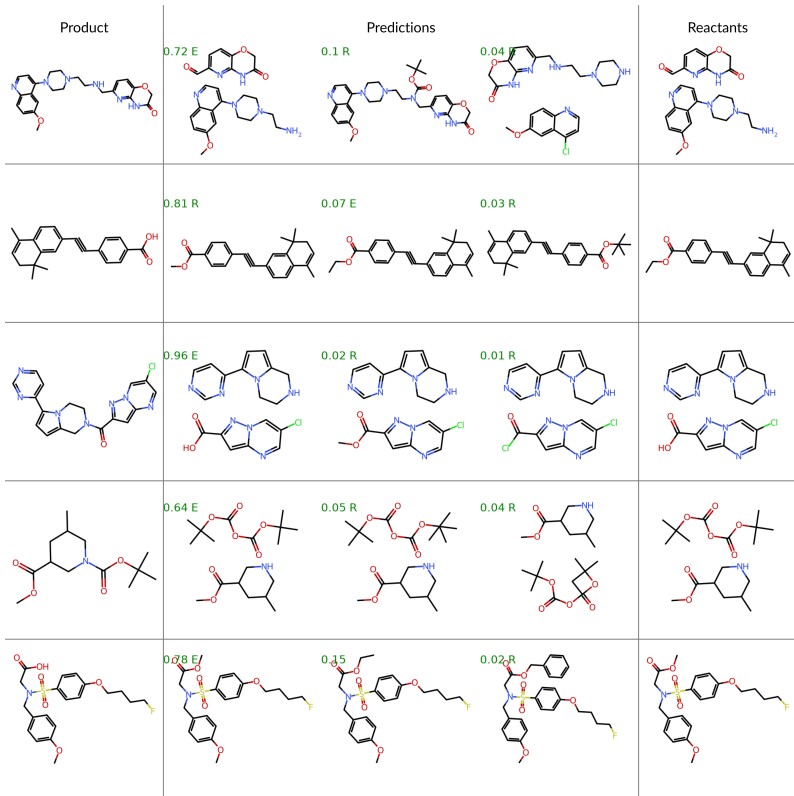

Figure 12: Visualizations of predictions made by RETRO PRODFLOW-RS. Examples are taken from the USPTO-50k test set randomly.

