# OpenReview forum: "RETRO SYNFLOW: Discrete Flow-Matching for Accurate and Diverse Single-Step Retrosynthesis"
_NeurIPS.cc/2025/Conference — NeurIPS 2025 poster_

### Official Review · Reviewer_L2PP · 2025-07-01

**Clarity:** 3
**Significance:** 3
**Originality:** 3
**Rating:** 4
**Confidence:** 4

**Summary:**

The paper presents RETRO PRODFLOW (RSF), a template‑free framework for single‑step retrosynthesis that is built on discrete flow matching. Two variants are introduced: 1) RSF learns a Markov bridge that maps product graphs directly to reactant graphs; 2) RSF first decomposes the product at a predicted reaction centre and then transports the resulting synthons to full reactants, treating the synthons as a more informative source distribution. At inference time the authors augment both variants with Feynman–Kac steering implemented through Sequential Monte Carlo; a forward‑synthesis model supplies the reward that preferentially resamples reactant sets likely to satisfy round‑trip consistency. On the standard USPTO‑50k benchmark the synthon‑based model reaches 60 % top‑1 exact‑match accuracy, roughly twenty percentage points above the best prior template‑free method, and improves top‑5 round‑trip accuracy by nineteen points over existing baselines, all while using 50 flow steps per sample.

**Questions:**

See Weaknesses

**Ethical Concerns:**

["NO or VERY MINOR ethics concerns only"]

**Limitations:**

yes.

**Quality:**

3

**Strengths And Weaknesses:**

Strengths:

+ The paper’s main contribution for casting retrosynthesis as a discrete flow‑matching problem is both conceptually clean and technically sound. By marginalising over conditional Markov paths, the authors obtain closed‑form probability velocities on categorical graph tokens, which allows fully likelihood‑based training instead of the surrogate objectives used in diffusion models. Employing synthons as an intermediate state is well motivated from a chemical standpoint and demonstrably boosts accuracy; the two‑stage design also makes the overall system more interpretable to practising chemists. Feynman–Kac steering is applied with care and yields a convincing diversity–feasibility trade‑off without extra training.

+ The manuscript is clearly written, the algorithmic details (transition kernels, reward definition, SMC schedule) are reproducible, and the ablations isolate the contribution of each component.

Weaknesses

- While the gains in exact‑match accuracy are impressive, the experimental study is limited to USPTO‑50k; it remains unclear how the approach scales to larger, noisier corpora or to multi‑product reactions.

- The work inherits a strong dependency on a separately trained reaction‑centre predictor whose errors are not quantified; the end‑to‑end robustness of the pipeline is therefore difficult to assess.

- No compute‑time comparison with diffusion‑based generators is provided, yet SMC resampling with four particles and fifty steps may be expensive in practice.

- Finally, the claim of being the first “flow‑matching framework for retrosynthesis” would benefit from a clearer differentiation from recent GFlowNet work that also learns stochastic policies over reaction graphs.

---

> ### Author Rebuttal · Authors · 2025-07-30
>
> We would like to thank the reviewer for the time and effort they spent on reviewing our work. We are glad that the reviewer found our work to be conceptually clean,  technically sound, and our use of synthons to be well motivated.  We now address the main questions raised by the reviewer.
>
> ## Benchmarking on other datasets
>
> > While the gains in exact‑match accuracy are impressive, the experimental study is limited to USPTO‑50k; it remains unclear how the approach scales to larger, noisier corpora or to multi‑product reactions.
>
> We thank the reviewer for their suggestion on testing on the USPTO Full and other datasets. We believe exploring how our methods scale to datasets such as USPTO Full is an interesting direction for future work. In particular, we follow the standard experimental practice in many prior recent works, including other diffusion-based approaches and synthon-based approaches, which solely report retrosynthesis results on the USPTO-50k dataset [2-8]. Furthermore, we have conducted comprehensive ablations on USPTO-50k, and we argue that our results are comprehensive in empirical caliber. Nevertheless, we will include a discussion on these points as part of our conclusion in our updated draft.
>
> ## GFlowNet vs Flow Matching
>
> > The claim of being the first “flow‑matching framework for retrosynthesis” would benefit from a clearer differentiation from the recent GFlowNet.
>
> We appreciate the reviewer’s observation and the opportunity to clarify how our approach differs from GFlowNets. GFlowNets and flow matching are fundamentally distinct frameworks. GFlowNets aim to sample trajectories whose probabilities are proportional to a given reward function. They model how probability mass flows through a directed acyclic graph (DAG) of partial trajectories. In contrast, flow matching models a continuous-time Markov chain that transports mass between arbitrary source and target distributions. Unlike GFlowNets, flow matching does not rely on a reward function to learn this mapping. Given these conceptual differences, we believe it is appropriate to state that our work is the first to apply flow matching to retrosynthesis.
>
> > The work inherits a strong dependency on a separately trained reaction‑centre predictor whose errors are not quantified; the end‑to‑end robustness of the pipeline is therefore difficult to assess.
>
> Although the two-stage framework relies on identifying synthons, this is not a problem in practice since reaction-center prediction can be performed with high accuracy. An off-the-shelf reaction center prediction model achieves a top-2 and top-3 accuracy of 85.1% and 89.5% respectively.  The more challenging aspect of retrosynthesis is synthon completion [9], which is the main aspect addressed by our work.
>
> ## Compute time Comparison
>
> The following table provides the average inference time to sample 100 reactants for a given product in the USPTO-50k test set. We benchmark on an RTX 3090 with 24 GB of memory.
>
> | Method | Mean time |
>  |   ----      |       ----       |
> |   GLN            | 0.295|
> | LocalRetro    | 0.022|
> | RetroBridge  |  56.0 |
> | RSF (K=1)    | 5.4    |
> | RSF (K=2)    | 15. 6 |
> | RSF (K=4)    |  30.6 |
> | RSF (K=6)    |  45.7 |
> | RSF (K=8)    |  53.9 |
>
> We note that 30 seconds for sampling a set of 100 reactants for a given product is a completely feasible time for applications of models like ours. In particular, this speed does not prevent the use of our method as a component of a multistep retrosynthesis planning pipeline. Additionally, the sampling time for RetroBridge with T=500 steps and 100 reactants is ~50 seconds.
>
> ## Concluding remarks
>
> We thank the reviewer again for their valuable feedback. We hope that our rebuttal addresses their questions and concerns, and we kindly ask the reviewer to consider a fresher evaluation of our paper if the reviewer is satisfied with our responses. We are also more than happy to answer any further questions that arise.
>
> ## References
>
> [1] Dai et al. “Retrosynthesis Prediction with Conditional Graph Logic Network” Neurips 2019.
>
> [2] Laabid et al. “Equivariant Denoisers Cannot Copy Graphs: Align Your Graph Diffusion Models”, ICLR 2025.
>
> [3] Igashov et al. “RetroBridge: Modeling Retrosynthesis with Markov Bridges”, ICLR 2024
> [4] Somnath et al. “Learning Graph Models for Retrosynthesis Prediction” NeurIPS 2021.
>
> [5] Shi et al. “A Graph to Graphs Framework for Retrosynthesis Prediction”, ICML 2021.
>
> [6] Zhong et al. “Retrosynthesis prediction using an end-to-end graph generative architecture for molecular graph editing”  2023.
>
> [7] Gianski et al. “RetroGFN: Diverse and Feasible Retrosynthesis using GFlowNets”, 2024.
>
> [8] Chen and Jung. “Deep Retrosynthetic Reaction Prediction using Local Reactivity and Global Attention.” 2021
>
> [9] Smith. “Organic Synthesis”, 2017.

---

### Official Review · Reviewer_5pkR · 2025-07-03

**Clarity:** 4
**Significance:** 3
**Originality:** 2
**Rating:** 4
**Confidence:** 3

**Summary:**

This paper introduces RETRO SYNFLOW (RSF), a template free, graph based framework for single step retrosynthesis built on discrete flow matching. The authors adapt Feynman–Kac (FK) steering with Sequential Monte Carlo resampling, guided by a forward synthesis reward that enforces round trip consistency, while enhancing the diversity and chemical feasibility of the results. Experimentally, RSF achieves a state-of-the-art top-1 accuracy of 60.0% on the USPTO-50k benchmark, outperforming prior template-free methods by 20%. The appendix details sensitivity to synthon allocation, number of SMC particles and sampling steps, and provides runtime statistics.

**Questions:**

* What is the scale of the model (eg number of parameters)? How does inference time scale with the number of particles?
* Have you tested on USPTO Full, MIT Mix or proprietary sets? A small scale study would strengthen the significance claim and could raise the quality score.
* To clarify, are synthons just the product graph with deleted edges? If so, why is it such a good inductive bias? How many edges are being deleted on average?
* Can you report the proportion of FK steered outputs that are actually chemically valid (validated with RDKit) but misscored by the oracle? Sensitivity analysis on oracle noise would clarify reliability
* Please provide inference wall time versus template based baselines, and discuss GPU memory needs for large batch SMC.
* RSF assumes the top 2 reaction center predictions; how does accuracy degrade if the correct center is ranked lower, and could joint training of center and flow alleviate this?

**Ethical Concerns:**

["NO or VERY MINOR ethics concerns only"]

**Final Justification:**

Happy with the author's responses and updated my score.

**Limitations:**

Yes

**Quality:**

3

**Strengths And Weaknesses:**

Strengths:
* The paper is clear, well written, and presents a novel application of discrete flow matching, exploiting its flexibility with respect to source and target distributions
* First application of discrete flow matching to retrosynthesis; For template-free methods, the paper demonstrates unequivocally strong results. The evaluations and ablations are straightforward and comprehensive
* The method will scale very well with more data and compute

Weakness:
* The methodological contribution is not very novel. Discrete flow matching with FK steering is a well-known method. Previous papers have also used synthons for retrosynthesis. Because the method is less novel, the expectation for state of the art results will be higher
* The advantage of RETRO SYNFLOW over template-based approaches is questionable. Local Retro is competitive and, in some cases, better. The authors claim that template free approaches are more flexible and generalize better, but it is unclear if better generalization shows up in the evaluation.
* The round trip metrics seem flawed because the method directly optimizes for them, whereas others do not. We can see this in the Figure 5 ablations, where round trip metrics increase with the number of particles, but accuracy does not. Round trip metrics appear to be proxies for quality, but do not have obvious direct applications.
* Only USPTO 50k is used; no larger or more diverse sets (e.g., USPTO Full, Pistachio)
* Molecular Transformer accuracy bounds the reward; its errors may bias results but are not quantified

---

> ### Author Rebuttal · Authors · 2025-07-30
>
> We thank the reviewer for the time and effort spent reading and engaging with our work. We are glad that the reviewer found our work to be a novel application of discrete flow matching, demonstrating strong results with comprehensive evaluations and ablations. We now address the reviewer's questions.
>
> ## Novelty of RetroSynFlow
>
> We acknowledge the reviewers' concern that the core novelty of RetroSynFlow may not be initially apparent. We would like to politely push back against this assertion, as discrete flow matching is not an established approach for single-step retrosynthesis. Consequently, adapting the discrete flow matching framework—whose primary success has been with masked/absorbing states—to a setting where the prior is a set of reactants/synthons is a simple yet effective novel design contribution of our work. Furthermore, to the best of our knowledge, the use of FK steering with a discrete flow matching model operating on synthons is also novel. In particular, our proposed use of the forward-synthesis model as a reward for inference-time steering is a novel insight that is specifically tailored to the single-step retrosynthesis setup. In contrast, prior work like RetroGFN [7]  had only considered reward models **during training**. We argue that these design decisions lead to an overall approach whose combination of components is novel and drives the current empirical success over prior SOTA template free approaches.
>
> ## RetroSynFlow vs. template-based approaches
>
> > The advantage of RETRO SYNFLOW over template-based approaches is questionable. Local Retro is competitive and, in some cases, better. The authors claim that template free approaches are more flexible and generalize better, but it is unclear if better generalization shows up in the evaluation.
>
> We appreciate the reviewer’s nuanced feedback on the distinction between template-free and template-based approaches. Template-based methods rely on a fixed library of templates extracted from datasets, which encode only the transformation rules observed in those datasets. Thus, these methods introduce a strong inductive bias, highlighted by the strong performance of LocalRetro, which outperforms many template-free baselines. However, this reliance on predefined templates limits the chemical space these methods can explore, making it difficult for them to generalize to unseen reactions [14–16]. For instance, GLN achieves top-1 accuracy of 52.5% on USPTO-50k but drops to 39.5% on USPTO-Full. Also, extracting reaction templates can be expensive, either requiring expert chemists to handcraft rules or relying on computationally intensive subgraph matching algorithms [17]. Automatically extracted template sets may contain errors, as they are often less curated and validated [18]. Finally, we note that RetroSynFlow outperforms existing template-free baselines, further demonstrating its effectiveness.
>
> ## Use of round-trip accuracy
>
> > The round trip metrics seem flawed because the method directly optimizes for them, whereas others do not. We can see this in the Figure 5 ablations, where round trip metrics increase with the number of particles, but accuracy does not. Round trip metrics appear to be proxies for quality, but do not have obvious direct applications.
>
> We use round-trip accuracy to address the limitations of exact-match accuracy, which overlooks the fact that multiple valid reactant sets can produce the same product but may not appear in the dataset. Top-$k$ round-trip accuracy captures this by measuring the proportion of feasible reactants (via a forward model) among the top-$k$ predictions. Importantly, it can be optimized independently of top-$k$ accuracy. Our reward-steered methods improve round-trip accuracy by generating diverse, feasible reactant sets while maintaining competitive top-$k$ performance by including ground-truth reactants among their outputs. Even without reward steering, our model performs on par with state-of-the-art methods on round-trip metrics. Although not a perfect measure, round-trip accuracy is widely used in the literature [1–3, 7–9, 11].
>
> ## Benchmarking on other datasets
>
> We thank the reviewer for their suggestion on testing on the USPTO Full and other datasets. We believe exploring how our methods scale to datasets such as USPTO Full is an interesting direction for future work. In particular, we follow the standard experimental practice in many prior recent works, including other diffusion-based approaches and synthon-based approaches, which solely report results on the USPTO-50k dataset [1-9]. Furthermore, we have conducted comprehensive ablations on USPTO-50k, and we argue that our results are comprehensive in empirical caliber. Nevertheless, we will include a discussion on these points as part of our conclusion in our updated draft
>
> ## Questions
> > What is # of parameters of the model? How does inference time scale with the number of particles? Please provide inference wall time versus template-based baselines, and discuss GPU memory needs for SMC.
>
> The model has 3.38 million parameters. The following table provides the average inference time to sample 100 reactants for a given product in the USPTO-50k test set. We benchmark on an RTX 3090 with 24 GB of memory.
>
> | Method | Mean time |
> |    ---       |         ---      |
> |   GLN            | 0.295|
> | LocalRetro    | 0.022|
> |RetroBridge   |  56.3 |
> | RSF (K=1)    | 5.4    |
> | RSF (K=2)    | 15. 6 |
> | RSF (K=4)    |  30.6 |
> | RSF (K=6)    |  45.7 |
> | RSF (K=8)    |  53.9 |
>
> We note that 30 seconds for sampling a set of 100 reactants for a given product is a completely feasible time for applications of models like ours. In particular, this speed does not prevent the use of our method as a component of a multistep retrosynthesis planning pipeline. Additionally, the sampling time for RetroBridge with T=500 steps and 100 reactants is ~50 seconds.
>
> >  Molecular Transformer's errors may bias results but are not quantified.
>
> We evaluate the results of Molecular Transformer on the USPTO-50k dataset and obtain the following results.
>
> |Top-1| Top-3| Top-5 | Top-10|
> |   ---    |   ---   |    ---   |      ---  |
> | 75.0 | 82.0  | 83.0   |   83.0 |
>
> We refer the reviewer to Appendix C, specifically Table 6 and Figures 8 and 9, for a brief discussion and visualization of the potential bias introduced by the forward-synthesis model. In particular, there are cases where the reward-steered model may generate reactants that are not feasible and do not match the ground truth, whereas the non-steered version produces a correct prediction.
>
> > To clarify, are synthons just the product graph with deleted edges? If so, why is it such a good inductive bias? How many edges are being deleted on average?
>
> Synthons correspond to the product graph with deleted or modified edges. In the USPTO-50k dataset, the reactions have one edge deleted on average.  Identifying synthons is a key step in how human chemists approach retrosynthetic analysis, as it helps them visualize how a complex molecule can be constructed from simpler, idealized fragments that point to the actual reactants required for synthesis. [12, 13]  Employing synthons provides more information about the reaction center and, along with the product context, simplifies the generation task by allowing the denoiser model to potentially “localize” the changes that need to be made to transform the current noisy graph into the reactant molecule graph.
>
> > Can you report the proportion of FK steered outputs that are actually chemically valid (validated with RDKit) but misscored by the oracle? A sensitivity analysis on oracle noise would clarify reliability.
>
> For the non-steering outputs, 25% of the chemically valid reactants were misscored by the oracle. For the FK-steered outputs, only 14% of the chemically valid reactants were misscored by the oracle.
>
> ## Concluding remarks
>
> We hope that our responses were sufficient in clarifying all the great questions asked by the reviewer. We thank the reviewer again for their time, and we politely encourage the reviewer to consider updating their score if they deem that our responses in this rebuttal, along with the new experiments, merit it.
>
> ## References
>
> [1] Wan et al. “Retroformer: Pushing the Limits of End-to-end Retrosynthesis Transformer”, ICML 2022.
>
> [2] Laabid et al. “Equivariant Denoisers Cannot Copy Graphs: Align Your Graph Diffusion Models”, ICLR 2025.
>
> [3] Igashov et al. “RetroBridge”, ICLR 2024
>
> [4] Somnath et al. “Learning Graph Models for Retrosynthesis Prediction” NeurIPS 2021.
>
> [5] Shi et al. “A Graph to Graphs Framework for Retrosynthesis Prediction”, ICML 2021.
>
> [6] Zhong et al. “Retrosynthesis prediction using an end-to-end graph generative architecture for molecular graph editing” 2023.
>
> [7] Gianski et al. “RetroGFN”, 2024.
>
> [8] Chen and Jung. “Deep Retrosynthetic Reaction Prediction using Local Reactivity and Global Attention.” 2021
>
> [9] Qiao et al. “Advancing Retrosynthesis with Retrieval-Augmented Graph Generation”. AAAI 2025.
>
> [10] Schwaller et al. “Molecular Transformer: A Model for Uncertainty-Calibrated Chemical Reaction Prediction”. 2019
>
> [11] Zeng et al. “Ualign: pushing the limit of template-free retrosynthesis prediction with unsupervised SMILES alignment”. 2024
>
> [12] Corey, “General Methods for the Construction of Complex Molecules”. 1967.
>
> [13] Warren, “Designing Organic Syntheses: A Programmed Introduction to the Synthon Approach”. 1978.
>
> [14] Yan et al. “RetroXpert”. 2020
>
> [15] Segler et al. “Modelling Chemical Reasoning to Predict and Invent Reactions”. 2016.
>
> [16] Thakkar et al. “Datasets and their influence on the development of computer assisted synthesis planning tools in the pharmaceutical domain”. 2020
>
> [17] Liu et al. “Retrosynthetic reaction prediction using neural sequence-to-sequence models”. 2017
>
> [18] Yao et al. “Node-Aligned Graph-to-Graph: Elevating Template-free Deep Learning Approaches in Single-Step Retrosynthesis. 2024.

---

> > ### Comment · Reviewer_5pkR · 2025-08-07
> > **Reply to Author Rebuttal**
> >
> > Thanks for the clarification and additional experiments. Could you also clarify the following two questions
> >
> > * To clarify, are synthons just the product graph with deleted edges? If so, why is it such a good inductive bias? How many edges are being deleted on average?
> > * RSF assumes the top 2 reaction center predictions; how does accuracy degrade if the correct center is ranked lower, and could joint training of center and flow alleviate this?

---

> > > ### Author Response · Authors · 2025-08-08
> > >
> > > We thank the reviewer for their time, effort, and engagement with us during the discussion phase.
> > >
> > > ## Clarifying role of synthons
> > >
> > > > To clarify, are synthons just the product graph with deleted edges? If so, why is it such a good inductive bias? How many edges are being deleted on average?
> > >
> > > Synthons can be thought of as the product graph with either deleted or modified edges. In the USPTO-50k dataset, the reactions have one edge deletion on average. The two-stage formulation with synthons aligns closely with how human chemists intuit single-step retrosynthesis, by breaking down a complex molecule into simpler fragments that point to the underlying reactants [2, 3]. Using synthons as an intermediate step introduces a useful inductive bias by narrowing the model’s focus to the reaction center. Combined with the product context, this makes the generation task more structured: it helps the denoiser model concentrate on the localized modifications needed to transform the current graph into the correct reactants. We politely remind the reviewer that we mentioned the definition of synthon and their inductive bias in the original rebuttal.
> > >
> > > ## Reaction centers
> > >
> > > > RSF assumes the top 2 reaction center predictions; how does accuracy degrade if the correct center is ranked lower, and could joint training of center and flow alleviate this?
> > >
> > > While the two-stage framework relies on identifying synthons, this is not a problem since reaction-center prediction can be performed with high accuracy. An off-the-shelf reaction center prediction model achieves a top-2 and top-3 accuracy of 85.1% and 89.5% respectively.  The more challenging aspect of retrosynthesis is synthon completion [1], which is the main aspect addressed by our work.
> > >
> > > In Table 5 of Appendix B, we present an ablation study on the USPTO-50K validation set where we vary the number of reactants generated for the top-ranked synthon prediction. We observe that incorporating the top-2 synthon predictions leads to strong performance, and that generating more reactants for the highest-scoring synthon is beneficial. For example, when generating 50 reactants from both synthons, the top-1 accuracy is 48.5, but generating 70 reactants from the first synthon prediction and 30 reactants from the second synthon prediction results in a top-1 accuracy of 58.0. This is consistent with the intuition that prioritizing the highest-ranked synthon, which is more likely to be correct, should lead to better performance.
> > > Moreover, the second row in the table below shows the performance of RetroSynFlow when generating ten additional reactants from the third-ranked synthon prediction. The first entry is from the main table.
> > >
> > > | Model |Top-1| Top-3| Top-5 | Top-10|
> > > | ---     | ---   |   ---     |    ---     |    --- |
> > > |  RSF | 60.0 | 77.9  |  82.7  | 85.3   |
> > > |  (K=3 synthon) | 60.5 | 78.2  | 83.7   |   86.9 |
> > >
> > > These results indicate that using the third-ranked synthon yields modest improvements in top-5 and top-10 accuracy. When using two synthons where the first is incorrect and the second is correct, the top-5 and top-10 accuracies decrease slightly to 77.5% and 80.9%, respectively. This reinforces the robustness of our approach, even when the correct reaction center is ranked lower; using the top-2 predictions is typically sufficient for high performance.
> > >
> > > Lastly, we acknowledge that jointly training the reaction center and flow models may offer further improvements. A flow-matching model that generates synthons directly could support a fully probabilistic formulation. Additionally, conditioning the flow model on multiple reaction center predictions could be a promising direction for enhancing performance, which we leave as future work. We will include a discussion of this in our conclusion in the updated draft of the paper.
> > >
> > > ## Concluding remarks
> > >
> > > We thank the reviewer again for their feedback that allowed us to strengthen our work with additional clarifications, experiments, and ablations which we will include in the updated paper. We also invite the reviewer to ask any other lingering questions they may have or to potentially consider a fresher evaluation of our paper with these rebuttal responses in context.
> > >
> > > ## References
> > >
> > > [1] Smith. “Organic Synthesis”, 2017.
> > >
> > > [2] Corey, “General Methods for the Construction of Complex Molecules”. 1967.
> > >
> > > [3] Warren, “Designing Organic Syntheses: A Programmed Introduction to the Synthon Approach”. 1978.

---

### Official Review · Reviewer_FqVg · 2025-07-04

**Clarity:** 4
**Significance:** 4
**Originality:** 4
**Rating:** 5
**Confidence:** 3

**Summary:**

The task is retrosynthesis, i.e. identification of a  synthesis route  and the class of architecture is that of flow-matching; The authors have considered a Markov bridge that functions to cover the target product and the reactants. They used intermediate structures of the flow, synthons, i.e. intermediate fragments. There is a reaction center predictor that maps the synthons to reactants.

**Questions:**

See detailed comments under Weaknesses. Each point can be considered a prompt/question and I am willing to adjust my review if addressed appropriately.

**Ethical Concerns:**

["NO or VERY MINOR ethics concerns only"]

**Final Justification:**

The authors have addressed all my concerns. I read all the other reviewers comments on the different rebuttals; I raised the scores.

**Limitations:**

Please consider discussing limitations or avenues for future research at the end of the paper.

**Paper Formatting Concerns:**

ok

**Quality:**

4

**Strengths And Weaknesses:**

strengths:
    While previously addressed using Markov Bridges [1], the paper models the transformation from product to reactants as a continuous-time Markov chain over the discrete space of atoms and bonds using discrete flow matching, to my knowledge, for the first time
    The authors address diversity in the context of retrosynthesis prediction and use FK-steering to improve round-trip accuracy at inference time.
Intro does a good job at setting up an open research question on the retrosynthesis; the math is very clear and together with figures explain well the two steps, i.e. 1) mapping products directly to reactants; 2) leverages synthons to facilitate the generation task. The appendix is very informative.

weaknesses:
    The empirical results are impressive and push the state of the art, however there are a few limitations that the authors should address:

 The authors repeatedly describe their methods, including RSF, as "template-free", which is somewhat misleading. RSF employs a two-stage process of reaction center identification followed by synthon completion, and I would encourage the authors to consider repositioning the model as a semi-template method. This would align the work more accurately with the established literature [2] and would more clearly highlight that the core component leading to the state-of-the-art results lies in the DFM-based synthon completion module.
The reward oracle for FK-steering is an imperfect forward synthesis model. How might false positives and false negatives from this reward model bias the steered generation process? A discussion of this limitation, and perhaps a small-scale analysis of the forward model's accuracy on a subset of the generated reactions, would add some nuance.
The finding in Table 3 that removing the product context improves synthon completion accuracy is counter-intuitive. Can the authors provide a hypothesis or an analysis to explain this? Does this point to a limitation in how the denoiser handles conditional information?
Minor: While not necessary for this rebuttal, I’d be curious to see how RetroBridge performs with the same number of sampling steps used by SynFlow.
Minor: Line 151 says “Since some atoms present in the product may not appear in the reactants”, it should be the other way around – atoms in the reactants might not be present in the product.



[1] RetroBridge: Modeling Retrosynthesis with Markov Bridges, Ilia Igashov and Arne Schneuing and Marwin Segler and Michael Bronstein and Bruno Correia, 2024, arxiv 2308.16212,

[2] Learning Graph Models for Retrosynthesis Prediction, Vignesh Ram Somnath and Charlotte Bunne and Connor W. Coley and Andreas Krause and Regina Barzilay, 2021, arxiv 2006.07038.

---

> ### Author Rebuttal · Authors · 2025-07-30
>
> We would like to thank the reviewer for the time and effort they spent engaging with our work. We appreciate that the reviewer finds our application of discrete flow matching to retrosynthesis novel and the use of FK-steering to improve round-trip accuracy a strength of our work. We now address the questions and concerns.
>
> ## Template-free vs semi-template
>
> > The authors repeatedly describe their methods, including RSF, as "template-free", which is somewhat misleading. I would encourage the authors to consider repositioning the model as a semi-template method.
>
> We appreciate the reviewer’s constructive feedback on the positioning of our work as a semi-template method. Semi-template approaches aim to combine the interpretability of template-based models with the flexibility and generalization ability of template-free methods. These methods typically operate in two stages: the first stage involves predicting the reaction center to identify synthons, and the second stage, which we focus on, completes the synthons into valid reactants. While some prior works (e.g., [2]) do not explicitly differentiate between semi-template and template-free methods, the consensus in the literature is to classify such two-stage frameworks as semi-template approaches [3]. In light of this, we will revise the introduction and related work sections to clarify the characteristics of semi-template methods and more accurately position our contribution within this framework.
>
> ## Using imperfect forward synthesis model as the reward oracle
>
> > The reward oracle for FK-steering is an imperfect forward synthesis model. A discussion of this limitation, and perhaps a small-scale analysis of the forward model's accuracy on a subset of the generated reactions, would add some nuance.
>
> This is indeed a fair point. Forward-synthesis is an easier task than retrosynthesis, and correspondingly, forward models such as Molecular Transformer [1] have strong performance. However, they are still subject to false positives and false negatives, which can potentially bias the steering process towards invalid reactants and limit performance in round-trip accuracy. We will add this to the limitation sections at the end of our paper.
>
> We evaluate the results of Molecular Transformer on the USPTO-50k dataset and obtain the following results.
>
> |Top-1| Top-3| Top-5 | Top-10|
> |   ---    |   ---   |    ---   |      ---  |
> | 75.0 | 82.0  | 83.0   |   83.0 |
>
> We refer the reviewer to Appendix C, specifically Table 6 and Figures 8 and 9, for a brief discussion and visualization of the potential bias introduced by the forward-synthesis model. In particular, there are cases where the reward-steered model may generate reactants that are not feasible and do not match the ground truth, whereas the non-steered version produces a correct prediction.
>
> Furthermore, for the non-steering outputs, 25% of the chemically valid reactants were misscored by the forward-model oracle. For the FK-steered outputs, only 14% of the chemically valid reactants were misscored by the oracle.
>
> ## Miscellaneous
> > The finding in Table 3 that removing the product context improves synthon completion accuracy is counter-intuitive.
>
> We apologize for the confusion regarding this typo. Indeed, using the product context increases accuracy, and we have fixed this typo so the columns are labeled correctly.
>
> ## Concluding remarks
>
> We hope that our responses were sufficient in clarifying all the great questions asked by the reviewer. We thank the reviewer again for their time, and we politely encourage the reviewer to consider updating their score if they deem that our responses in this rebuttal, along with the new experiments, merit it.
>
> ## References
>
> [1] Schwaller et al. “Molecular Transformer: A Model for Uncertainty-Calibrated Chemical Reaction Prediction” 2025.
>
> [2] Igashov et al. “RetroBridge: Modeling Retrosynthesis with Markov Bridges”. 2024
>
> [3] Somnath et al. “Learning Graph Models for Retrosynthesis Prediction”. 2021

---

> > ### Comment · Reviewer_FqVg · 2025-08-07
> >
> > I am happy of the clarifications and I increase my marks
> >
> > Quality: 4: good
> > Clarity: 4: good
> > Significance: 4: good
> > Originality: 4: good

---

### Official Review · Reviewer_R1w9 · 2025-07-13

**Clarity:** 3
**Significance:** 4
**Originality:** 4
**Rating:** 5
**Confidence:** 5

**Summary:**

The authors use discrete flow matching for the problem of retrosynthesis prediction, and use also Feynman Kac steering with a forward reaction prediction model.

This is as far as I know the first use of flow  matching and FK steering for this task. Nice!

The model achieves good performance.


Overall this paper should be accepted to the conference after some comments have been addressed

**Questions:**

1) which Codebase have the authors used to compute the metrics?

**Ethical Concerns:**

["NO or VERY MINOR ethics concerns only"]

**Final Justification:**

updated score based on discussion.

one thing to note is that retrobridge code does not perform stereochemically aware assignment, for the final version my recommendation would be to run the metrics through www.github.com/microsoft/syntheseus

**Limitations:**

Yes

**Quality:**

3

**Strengths And Weaknesses:**

Strengths:
Interesting model formulation and results.
Nice use of forward model via FK steering

Weaknesses:
The FK ablation isn’t clear. Most Other baselines in the main results table don’t use the forward model as additional filtering, thus the comparison isn’t fully fair. I suggest to add the results without FK steering to the main table as well

Missing citations to highly relevant work
https://arxiv.org/abs/2405.17656

Other works that have used forward model steering/filtering for synthesis planning:
- Segler er al Nature 2018
- Coley et al Science 2019
- Schwaller et al Chemical Science 2020

The chimera model by Maziarz er al https://arxiv.org/pdf/2412.05269 outperforms retrosynflow for all but top1 accuracy, even without an forward model filters or steering (see Extended Data Table1). This should be added to the main results table.


Comment:
This reviewer will increase score after discussion of above points

---

> ### Author Rebuttal · Authors · 2025-07-30
>
> We would like to thank the reviewer for the time and effort they spent on reviewing our work!  We appreciate that the reviewer found our model formulation and use of the forward model for FK steering to be interesting. We now address the main points raised in the review.
>
> ## Ablation without FK steering
>
> We acknowledge the reviewer's comment that our ablation should include a result that does not make use of FK steering. We believe that there may be a slight confusion, which we now hope to clarify. In our main results, we report our initial model, RetroProdFlow, which directly transports product molecules to their corresponding reactants, and importantly, is devoid of any FK steering. As evidenced by Table 1 results, we find RetroProdFlow to be a performant model on par with previous template-free methods like RetroBridge [6], but short in performance compared to RetroSynFlow, which is our primary model. Consequently, we argue that the baseline that the reviewer requests is already present in the main paper.
>
> In addition, we further argue that our proposed approach RetroSynFlow, is a combination of flow matching with the distribution of synthons serving as a strong inductive prior and FK steering to facilitate improvements in round-trip accuracy. Therefore, it is natural to consider these two components in one system rather than in isolation.
>
> We hope that this clarification here fully alleviates the reviewers' valid concern, and we will update the wording in the main paper to highlight that RetroProdFlow does not employ FK steering.
>
> ## Citations to related work
>
> We thank the reviewer for pointing us to these related works. In our updated draft, we will include [1,2,3] in our related works section to discuss the use of a forward-synthesis model to filter reactants in retrosynthesis.  In particular, a key difference between our proposed RetroSynFlow and these related works is the use of a forward-synthesis model to aid in multi-step retrosynthesis planning, while our work pertains to **single-step retrosynthesis prediction**. Consequently, these works do not report results on the USPTO-50k dataset, which we use in our evaluation.
>
> With regards to related works [4]  to the related works section and [5] to our baselines. The key differences between our work and [4] are that the denoiser model in [4] requires the atom-mapping as input, while our flow-matching denoiser does not. Furthermore, [4] employs absorbing state diffusion, which assumes that the source distribution is an absorbing state. In stark contrast, discrete flow matching allows us to learn a map between synthons/products and reactants directly.
>
> With regards to Chimera [5], we argue that this is a framework that ensembles many different retrosynthesis models across both graph and SMILE string-based methods. As a result, our method is complementary to their framework as it can be employed in the ensemble. Additionally, since Chimera ensembles various models, it may potentially be more resource-intensive than our single method in RetroSynFlow.  We update the main table with following new entries:
>
> | Method | Top-1 | Top-3 | Top-5 | Top-10 |
>  ---           |   ----   |     ----  |   ----  |  ----- |
> | Chimera[5]  | 59.6 | 82.8 | 89.2 | 94.2 |
>
> ## Questions
> > Which Codebase have the authors used to compute the metrics?
>
> We used the evaluation code from RetroBridge [6] to compute metrics.
>
> ## Concluding remarks
>
> We thank the reviewer for their time and effort in reviewing our paper. We hope that our rebuttal here answers all the great points raised by the reviewer, allowing them to potentially upgrade their score as they initially suggested. We are also more than happy to answer any further questions that arise in the rebuttal period. Please let us know.
>
> ## References
>
> [1] Segler et al. “Planning chemical syntheses with deep neural networks and symbolic AI”, 2018.
>
> [2] Coley et al. “A robotic platform for flow synthesis of organic compounds informed by AI planning”, 2019.
>
> [3] Schwaller et al. “Predicting retrosynthetic pathways using transformer-based models and a hyper-graph exploration strategy”, 2020.
>
> [4] Laabid et al. “Equivariant Denoisers Cannot Copy Graphs: Align Your Graph Diffusion Models”, 2025.
>
> [5] Maziarz et al. “Chimera: Accurate retrosynthesis prediction by ensembling models with diverse inductive biases”, 2024.
>
> [6] Igashov et al. “RetroBridge: Modeling Retrosynthesis with Markov Bridges”. 2024

---

> > ### Comment · Reviewer_R1w9 · 2025-08-08
> >
> > updated score based on discussion.
> >
> > one thing to note is that retrobridge code does not perform stereochemically aware assignment, for the final version my recommendation would be to run the metrics through www.github.com/microsoft/syntheseus

---

### Author Response · Authors · 2025-08-06

Dear Reviewers,

Thank you again for your thoughtful reviews and constructive feedback. We hope our rebuttal has addressed your main concerns, and we’re happy to further clarify any points. Please feel free to leave additional questions or comments; we’re eager to engage and respond.

Sincerely,
Authors

---

### Note · Authors · 2025-08-12

We are glad that the reviewers found our work well-written and the first to employ Discrete Flow Matching for retrosynthesis, with comprehensive ablations.

### Novelty and Prior Works
We addressed Reviewer 5pKR’s concern about novelty by noting that discrete flow matching is not an established approach for retrosynthesis, and adapting it (traditionally used with masked states) to a setting where the prior is synthons is an effective contribution. Using the forward model reward oracle for inference-time steering is also novel. This combination of components drives the current empirical success over prior template-free methods. We updated our baselines with Chimera, an ensemble of diverse retrosynthesis methods, and argued that SynFlow is complementary and can be integrated into such a setup. We also clarified our FK steering ablation by noting that RetroProdFlow does not employ FK steering; since the novelty of RetroSynFlow derives from combining synthon flow matching with FK steering, it is natural to consider these components together. After these discussions, Reviewer R1w9 and FqvG upgraded their scores.

### Large Datasets
Reviewers 5pKR and L2PP raised concerns about the scope of the USPTO-50k dataset. However, it is standard practice, followed by at least ten recent works, including diffusion and synthon-based methods, to report results solely on USPTO-50k. Our comprehensive ablations further demonstrate the empirical rigor of our study.

### Ablations
Reviewers 5pKR and FqVg questioned the errors of the forward-synthesis model and the round-trip accuracy metric. We showed that forward synthesis is easier than retrosynthesis, and forward models achieve high accuracy. Round-trip accuracy is widely used, and only 14% of FK-steered outputs were misscored by the oracle. While the two-stage framework depends on identifying synthons, off-the-shelf reaction-center prediction achieves 85.1% top-2 accuracy; we focus on the harder synthon completion task.

### Conclusion
We politely note that Reviewer 5pKR did not reply to our comments. We addressed their concerns by clarifying novelty, demonstrating the scope of our evaluation, reinforcing the validity of the forward and reaction-center models, and providing compute-time comparisons. Given the breadth of our responses, we believe our rebuttal convincingly addresses their points. We appreciate the constructive discussions, which resolved all outstanding concerns and led to improved reviewer scores.

---

### Decision · Program_Chairs · 2025-09-17

**Decision:**

Accept (poster)

**Comment:**

The paper is the first to apply discrete flow matching to retrosynthetic prediction.

The model achieves a strong performance on the USPTO50k benchmark. LocalRetro outperforms the method for larger K. The comparison on roundtrip shows state-of-the-art performance, but not all method used forward model based fine-tuning (which is part of the metric).

Experiments were conducted on USPTO50k, the standard testbed for the subfield, but not fully realistic or representative of practical applications. It remains to be seen how the method scales to larger datasets.

Three reviewers recommended acceptance, while one recommended rejection of the work.

The impact of the work is at the moment not fully clear, given the limited experimental scope and non-state-of-the-art performance compared to methods such as LocalRetro. However, all in all, it is a solid demonstration of the applicability of discrete flows to retrosynthetic prediction, which warrants acceptance.

Last but not least, I would like to ask politely the Authors to clarify in the abstract the statement "Empirically, we demonstrate RSF achieves 60.0% top-1 accuracy, which outperforms the previous SOTA by 20%”, which doesn’t seem to be the case given Table 1.